# Can Long-Context Language Models Subsume Retrieval, SQL, and More?

## Abstract

Long-context language models (LCLMs) have the potential to revolutionize our approach to tasks traditionally reliant on external tools like retrieval systems or databases. Leveraging LCLMs' ability to natively ingest and process entire corpora of information offers numerous advantages. It enhances user-friendliness by eliminating the need for specialized knowledge of tools, provides robust end-to-end modeling that minimizes cascading errors in complex pipelines, and allows for the application of sophisticated prompting techniques across the entire system. To assess this paradigm shift, we introduce LOFT, a benchmark comprising of real-world tasks requiring context up to millions of tokens designed to evaluate LCLMs' performance on in-context retrieval and reasoning. Our findings reveal that LCLMs can already achieve textual, visual, and audio retrieval performance comparable to specialized systems such as Gecko and CLIP, while still facing challenges in areas like multi-hop compositional reasoning required in SQL-like tasks. Notably, prompting strategies significantly influence performance, emphasizing the need for continued research as context lengths grow. Overall, LOFT provides a rigorous testing ground for LCLMs, showcasing their potential to supplant existing paradigms and tackle novel tasks as model capabilities scale.[1]

## 1 Introduction

Long-context language models (LCLMs) [42, 35, 4, 8] hold the promise of reshaping artificial intelligence by enabling entirely new tasks and applications while eliminating the reliance on tools and complex pipelines previously necessary due to context length limitations [17, 26]. By consolidating complex pipelines into a unified model, LCLMs ameliorate issues like cascading errors [7] and cumbersome optimization [23, 48], offering a streamlined end-to-end approach to model development. Moreover, techniques such as adding instructions [21, 46, 11], incorporating few-shot examples [9], and leveraging demonstrations via chain-of-thought prompting [34, 47] can be seamlessly integrated to optimize LCLMs for the task at hand.

However, realizing the full potential of LCLMs necessitates rigorous evaluation on truly long-context tasks useful in real-world applications. Existing benchmarks fall short in this regard, relying on synthetic tasks like the popular "needle-in-haystack" [19, 25] or fixed-length datasets that fail to keep pace with the evolving definition of "long-context" [6]. Critically, existing evaluations do not adequately stress-test LCLMs on these paradigm-shifting tasks.

To address this, we introduce the **Lo**ng-Context **F**ron**t**iers (LOFT) benchmark, a suite of six tasks comprising over 35 datasets spanning text, visual, and audio modalities designed to push LCLMs to their limits and gauge their real-world impact. Unlike previous benchmarks, LOFT allows for

---

[1]We will publicly release our dataset and evaluation code upon acceptance.

Submitted to 38th Conference on Neural Information Processing Systems (NeurIPS 2024). Do not distribute.

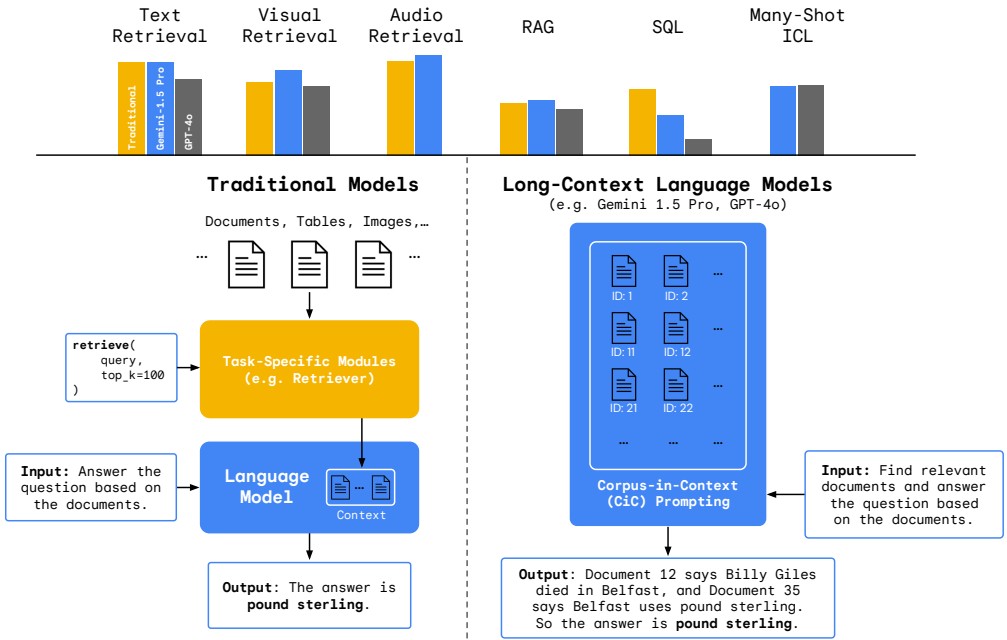

Figure 1: An overview of the LOFT benchmark, made of six tasks which measure LCLMs' ability to do in-context retrieval, reasoning, and many-shot learning on corpora up to millions of tokens. We compare the performance of LCLMs against traditional task-specific models (*e.g.,* CLIP for visual retrieval), which often rely on complex task-specific pipelines. Unlike traditional models, we show how LCLMs can simplify various tasks through Corpus-in-Context Prompting (Section 3).

automatic creation of varied context lengths, up to and exceeding 1 million tokens, ensuring rigorous evaluation as LCLMs continue to scale. Our benchmark focuses on the following areas where LCLMs have the potential for disruption:

- **Retrieval**: LCLMs can directly ingest and retrieve information from a corpus, eliminating the need for separate dual-encoder models [20, 33, 24, 37]. This addresses the information bottleneck found in retrievers [38] by enabling fine-grained interactions between query and corpus. We assess retrieval performance across text, visual, and audio modalities.

- **Retrieval-Augmented Generation (RAG)**: LCLMs simplify RAG pipelines by directly reasoning over a corpus, overcoming challenges like query decomposition [36] and mitigating cascading errors due to retrieval misses [7, 30].

- **SQL**: We explore LCLMs' capacity to process entire databases as text, enabling natural language database querying and bypassing conversion to a formal query language like SQL [53]. This potentially enables more expressive querying and handling of noisy or mixed-structured data.

- **Many-Shot ICL**: LCLMs can scale the number of examples from the tens in the traditional in-context learning setup to hundreds or thousands, removing the need to find the optimal set of few-shot examples to use [31].

The LOFT benchmark opens up a novel line of research on long-context prompting, which we introduce as Corpus-in-Context (CiC) Prompting (Section 3). Using this approach, we evaluate Gemini 1.5 Pro [Reid et al., 2024] and GPT-4o [Achiam et al., 2023] on LOFT. Figure 1 summarizes the performance of these LCLMs and traditional models on each task, showcasing how LCLMs can tackle LOFT tasks without specialized pipelines.

Our evaluation of state-of-the-art LCLMs on LOFT reveals several notable findings. At the 128k token level, the largest size comparable across all models, all closely match the performance of specialized systems in textual retrieval, with Gemini also performing significantly better than specialized systems in visual and audio retrieval. On complex multi-hop compositional reasoning tasks, however, all LCLMs lag considerably, highlighting significant room for improvement. Furthermore, rigorous ablations on prompting strategies such as the format of the corpora, the incorporation of

| | Dataset | Description | Avg. Cand. Length | # Cand. (128k) | Candidates | Input | Target |
|---|---|---|---|---|---|---|---|
| **Text Retrieval** | ArguAna | Argument Retrieval | 196 | 531 | Passages | Query | Passage ID(s) |
| | FEVER | Fact Checking | 176 | 588 | | | |
| | FIQA | Question Answering | 196 | 531 | | | |
| | MSMarco | Web Search | 77 | 1,174 | | | |
| | NQ | Question Answering | 110 | 883 | | | |
| | Quora | Duplication Detection | 14 | 3,306 | | | |
| | SciFact | Citation Prediction | 301 | 357 | | | |
| | Touché-2020 | Argument Retrieval | 330 | 329 | | | |
| | TopiOCQA | Multi-turn QA | 149 | 680 | | | |
| | HotPotQA | Multi-hop QA | 74 | 1,222 | | | |
| | MuSiQue | Multi-hop QA | 120 | 824 | | | |
| | QAMPARI | Multi-target QA | 132 | 755 | | | |
| | QUEST | Multi-target QA | 328 | 328 | | | |
| **Visual Retrieval** | Flickr30k | Image Retrieval | 258 | 440 | Images | Text Query | Image ID |
| | MS COCO | Image Retrieval | 258 | 440 | Images | Text Query | Image ID |
| | OVEN | Image-text Retrieval | 278 | 448 | Images+Texts | Image+Text Query | Wikipedia ID |
| | MSR-VTT | Video Retrieval | 774 | 140 | Videos | Text Query | Video ID |
| **Audio Retrieval** | FLEURS-en | | 249 | 428 | Speech | Text Query | Speech ID |
| | FLEURS-es | | 315 | 343 | | | |
| | FLEURS-fr | Audio Retrieval | 259 | 412 | | | |
| | FLEURS-hi | | 292 | 369 | | | |
| | FLEURS-zh | | 291 | 370 | | | |
| **RAG** | NQ | Question Answering | 110 | 883 | Passages | Question | Answer(s) |
| | TopiOCQA | Multi-turn QA | 149 | 680 | | | |
| | HotPotQA | Multi-hop QA | 74 | 1,222 | | | |
| | MuSiQue | Multi-hop QA | 120 | 824 | | | |
| | QAMPARI | Multi-target QA | 132 | 755 | | | |
| | QUEST | Multi-target QA | 328 | 328 | | | |
| **SQL** | Spider | Single-turn SQL | 111k | 1 | SQL Database | Question | Answer |
| | SParC | Multi-turn SQL | 111k | 1 | | | |
| **Many-Shot ICL** | BBH-date | Multiple-choice QA | 131 | 150 | Training Examples | Question | Answer |
| | BBH-salient | Multiple-choice QA | 246 | 104 | | | |
| | BBH-tracking7 | Multiple-choice QA | 205 | 123 | | | |
| | BBH-web | Multiple-choice QA | 43 | 150 | | | |
| | LIB-dialogue | Classification | 266 | 284 | | | |

Table 1: Tasks and datasets in the LOFT benchmark. LOFT has 6 types of tasks, 4 modalities, and 35 datasets in total. For each dataset, we show the average length of the candidates (Avg. Cand. Length) as well as the number of candidates (# Cand) in the 128k version of LOFT.

chain-of-thought reasoning, and the location of the target information within the context, reveal large variance in performance, underscoring the need for further research to make LCLMs robust and instructable. Taken together, our results on LOFT demonstrate that LCLMs can match the performance of many specialized systems, while also revealing ample headroom for improvement in robust long-context reasoning as context windows continue to scale.

# 2 LOFT: A 1 Million+ Token Long-Context Benchmark

The LOFT benchmark aims to cover a wide range of real-world applications where LCLMs can be employed. These tasks span from retrieving relevant documents for a query to extracting compositional information from databases. Table 1 lists all tasks and their corresponding datasets.

For each dataset in all tasks, we sample up to 100 test queries, 5 few-shot queries, and 10 development queries. To test how LCLMs perform with a larger number of tokens in their context, we create LOFT with four different length limits, namely 32k[2], 128k, 200k, and 1M. To allow testing the same set of queries over different context lengths, we process each dataset to have the same evaluation queries across different context lengths (except for SQL, where we split queries by database size).

**Retrieval & RAG** We include diverse text retrieval and RAG datasets, covering heterougenous retrieval tasks from BEIR [43], multi-turn conversational QA, multi-hop QA [49, 44], as well as multi-target QA that require set-operations [3, 32]. For retrieval, we also include multimodal datasets, covering image, video, and audio.

---

[2] Since the gold documents of 100 test queries alone often exceed 32k tokens, we do not include test queries for the 32k version. We report the development set performance for 32k instead.

All queries in each retrieval and RAG dataset shares a single corpus, mimicking real retrieval applications. To create this shared corpus, we first include all gold passages from few-shot, development, and test queries, and then randomly add random passages until reaching the desired context size (Figure 2). This construction ensures smaller corpora (e.g., 128k) are subsets of larger ones (e.g., 200k). Gold and random passages are shuffled to avoid positional biases. For fair comparison, our results comparing traditional baselines to LCLMs are also done on this same corpora of data.

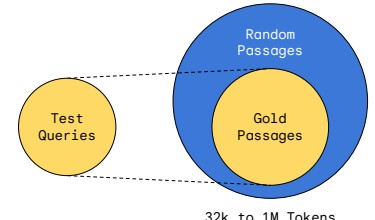

Figure 2: **Corpus creation** for retrieval and RAG. Given a set of test queries, we use their associated gold passages and other random passages to form the corpus.

**Many-shot ICL** We adapt datasets from Big Bench Hard (BBH) [40] and LongICLBench (LIB) [28] to evaluate LCLMs' many-shot in-context learning (ICL) capabilities. Similar to retrieval and RAG, we construct shared many-shot ICL contexts, ensuring training examples in smaller contexts are included in larger ones. Since all datasets are classification tasks, we guarantee that each classe is represented at least once.

**SQL** We evaluate SQL-like reasoning on Spider, a single-turn text-to-SQL dataset [51], and SparC, its multi-turn variant [52]. The corpus for each query is its associated database of one or more tables. For a maximum corpus size of $N$, we select queries with the largest databases still under $N$. Therefore, unlike shared corpus tasks, the query sets differ across LOFT sizes.

Given a maximum context length of $N \in \{32k, 128k, 200k, 1M\}$, we create a corpus up to a size of $0.9N$, to account for differences in tokenizers and reserving room for for instructions and formatting as we will see in Figure 3. Please refer to Appendix A for more details about dataset selection.

# 3 Corpus-in-Context Prompting

Traditionally, utilizing large corpora of passages, data tables, or training examples required specialized recipes or systems. Long-context language models (LCLMs) now enable direct ingestion and processing of entire corpora within their context window. This unlocks a novel prompting-based approach for solving , which we call **Corpus-in-Context** prompting (**CiC**, pronounced "sick").

## 3.1 Prompt Design

CiC prompting effectively combines established prompting strategies, tailoring them to leverage the unique capabilities of LCLMs for learning, retrieving and reasoning over in-context corpora. Figure 3 illustrates our key design choices, whose effectiveness is rigorously evaluated through extensive ablation studies in Section 5.

**Instructions** We first provide task-specific instructions to guide the LCLM's behaviors [21, 46, 11]. As an example for the retrieval task in Figure 3, we ask the model to read the corpus carefully and find relevant documents to answer the question.

**Corpus Formatting** We then insert the entire corpus into the prompt. The structure of the corpus significantly impacts retrieval performance. We find that careful formatting, such as repeating document IDs after passage text in retrieval, mitigates the effects of causal attention in decoder-only LCLMs, enhancing retrieval accuracy.

**Few-Shot Examples** Providing a limited number of demonstrations helps the LCLM grasp the desired response format and improves task accuracy [9]. Unlike common approaches where few-shot examples are independent, we ground all examples to the same corpus, aiming to teach the model understand the specific corpus. As we will see, positioning these examples can guide the model's attention to areas where it is typically weaker, mitigating "dead zones" in attention distribution.

Each few-shot example is accompanied by a Chain-of-Thought reasoning [34, 47]. We find adding Chain-of-Thought reasoning chains leads to the greatest benefits on tasks requiring complex multi-hop compositional reasoning.

**Query Formatting** The final evaluation query is formatted similar to each few-shot example (if any). Based on our query formatting, LCLMs complete the generation and provide answers.

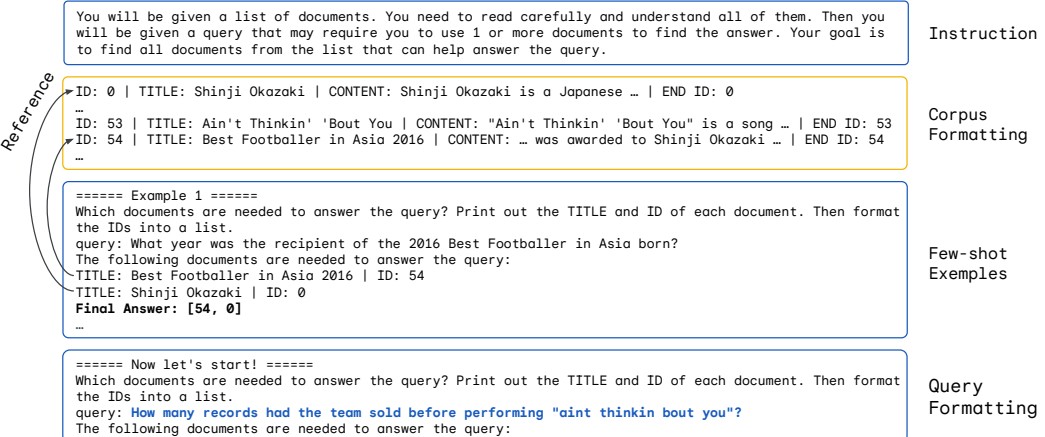

Figure 3: **Example of Corpus-in-Context Prompting** for retrieval. CiC prompting leverages large language models' capacity to follow instructions, leverage few-shot exemplars, and benefit from reasoning demonstrations to retrieve and reason over large corpora provided in context.

## 3.2 Discussion on Efficiency

Encoding a one million token context can be slow and computationally expensive. One key advantage of CiC prompting is its compatibility with prefix-caching in autoregressive language models as the query appears at the end of the prompt. This means *the corpus only needs to be encoded once*, similar to the indexing process in traditional information retrieval or database systems.

## 4 LOFT Tasks and Primary Results

Our evaluation on LOFT employs two state-of-the-art LCLMs: Google's **Gemini-1.5-Pro** [42] and OpenAI's **GPT-4o** [35]. These models were selected because their APIs support the most modalities in the benchmark. Their maximum context lengths are 2 million and 128k tokens, respectively. We use their official APIs[3],[4] for the evaluation. A small number of API calls were blocked due to various reasons, which were treated as incorrect.

### 4.1 Text Retrieval

We adopt Gecko [24], a state-of-the-art retriever as the tradi-
tional task-specific baseline. Gecko is a dual-encoder model
fine-tuned on extensive text retrieval and similarity tasks. To
ensure fair comparison, we use the same corpus used to test the
LCLMsto evaluate Gecko.

**Results**   Results in Table 2 demonstrate that at 128k context,
Gemini-1.5-Pro perform comparably to Gecko. This is notable,
as LCLMs have not undergone specialized contrastive learning
for retrieval. While LCLMs's performance does degrade when
scaling the corpus to millions of tokens (Figure 5), this initial
parity suggests the potential of LCLMs for retrieval tasks.

**Positional Analysis**   To better understand the cause of perfor-
mance degradation of LCLMs on larger context length datasets,
we investigate how the position of gold and few-shot documents
in the corpus influences retrieval [29].

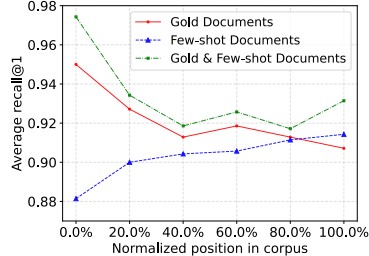

Figure 4: **Positional Analysis**.
We vary the position of gold and
few-shot documents within the cor-
pus (0% = beginning, 100% = end).

Figure 4 reveals that performance drops as gold documents move towards the end of the corpus, suggesting reduced attention in later sections. Conversely, placing few-shot examples at the end improves recall, indicating their ability to mitigate attention weaknesses in this region. Co-locating gold and few-shot documents consistently boosts performance. This demonstrates how few-shot

---
[3]https://ai.google.dev/gemini-api
[4]https://platform.openai.com/docs/models/gpt-4o

| | Dataset | Gemini-1.5$_{Pro}$ | GPT-4o | Traditional |
|---|---|---|---|---|
| **Text Retrieval** | ArguAna | 0.84 | 0.54 | 0.75 |
| | FEVER | 0.98 | 0.92 | 0.97 |
| | FIQA | 0.79 | 0.20 | 0.83 |
| | MSMarco | 0.95 | 0.89 | 0.97 |
| | NQ | 1.00 | 0.92 | 0.99 |
| | Quora | 0.93 | 0.95 | 1.00 |
| | SciFact | 0.88 | 0.80 | 0.85 |
| | TopiOCQA | 0.31 | 0.29 | - |
| | Webis-2020 | 0.91 | 0.71 | 0.88 |
| | HotPotQA$^\dagger$ | 0.90 | 0.70 | 0.92 |
| | MuSiQue$^\dagger$ | 0.42 | 0.18 | 0.29 |
| | QAMPARI$^\dagger$ | 0.61 | 0.21 | 0.57 |
| | QUEST$^\dagger$ | 0.30 | 0.19 | 0.54 |
| | **Average$^\ddagger$** | **0.91** | 0.74 | **0.91** |
| **Visual Retrieval** | Flickr30k | 0.84 | 0.65 | 0.75 |
| | MS COCO | 0.77 | 0.44 | 0.66 |
| | MSR-VTT | 0.76 | 0.72 | 0.64 |
| | OVEN | 0.93 | 0.89 | 0.79 |
| | **Average** | **0.83** | 0.68 | 0.71 |
| **Audio Retrieval** | FLEURS-en | 1.00 | - | 0.98 |
| | FLEURS-es | 0.99 | - | 0.99 |
| | FLEURS-fr | 1.00 | - | 1.00 |
| | FLEURS-hi | 1.00 | - | 0.74 |
| | FLEURS-zh | 1.00 | - | 1.00 |
| | **Average** | **1.00** | - | 0.94 |
| **RAG** | HotPotQA | 0.72 | 0.76 | 0.61 |
| | MuSiQue | 0.53 | 0.48 | 0.45 |
| | NQ | 0.81 | 0.76 | 0.70 |
| | QAMPARI | 0.39 | 0.20 | 0.51 |
| | QUEST | 0.28 | 0.12 | 0.31 |
| | TopiOCQA | 0.34 | 0.28 | - |
| | **Average$^\ddagger$** | **0.55** | 0.46 | 0.51 |
| **SQL** | Spider | 0.40 | 0.14 | 0.74 |
| | SParC | 0.36 | 0.13 | 0.55 |
| | **Average** | 0.38 | 0.14 | **0.65** |
| **Many-Shot ICL** | BBH-date | 0.88 | 0.81 | - |
| | BBH-salient | 0.78 | 0.64 | - |
| | BBH-tracking7 | 0.33 | 0.81 | - |
| | BBH-web | 0.67 | 0.57 | - |
| | LIB-dialogue | 0.76 | 0.67 | - |
| | **Average** | 0.68 | **0.70** | - |

Table 2: **Main Results on LOFT 128k context test set**. We show performances of two LCLMs (Gemini-1.5$_{Pro}$ and GPT-4o) as well as baselines that are traditionally used to solve these tasks. For the evaluation metrics: text, visual, and audio retrieval use Recall@1; RAG uses span-level exact match; SQL uses execution accuracy; and many-shot prompting uses accuracy. $^\dagger$: retrieval datasets with multiple gold targets use mRecall@$k$ (Appendix A). $^\ddagger$: The average text retrieval and RAG performance excludes TopiOCQA as the traditional baseline does not support multi-turn queries.

162 examples can strategically counterbalance areas of weak attention, offering a promising approach to
163 overcome performance degradation in large corpora. Per-dataset analysis is provided in Appendix C.

## 4.2 Visual Retrieval

165 We employ CLIP-L/14, a widely used text-to-image retrieval model, as our traditional task-specific
166 baseline [37]. For Flickr30k and MSCOCO, CLIP performs text-to-image retrieval. For MSR-VTT,
167 it performs text-to-video retrieval by averaging scores across frames. For OVEN, due to the lack of
168 suitable open-source image-to-text models, we approximate image-to-text retrieval also using CLIP's
169 text-to-image retrieval.

170 **Results** Gemini 1.5 Pro outperforms GPT-4o across all four visual benchmarks (Table 2). Notably,
171 as shown in Figure 5, Gemini 1.5 Pro maintains a performance advantage over the CLIP across all
172 visual benchmarks and context lengths.

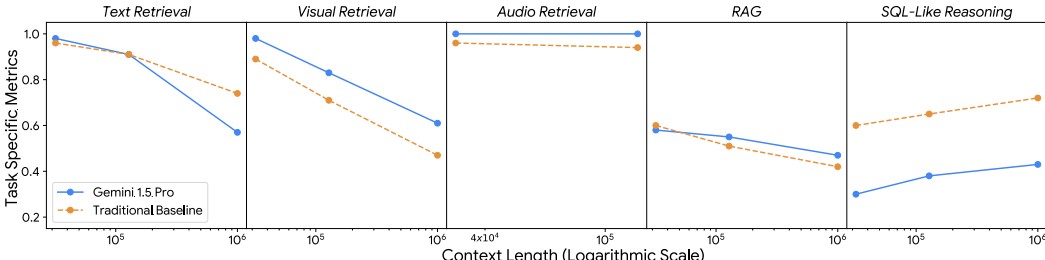

Figure 5: **Scaling results** of LCLMs compared to a traditional baseline by scaling the corpus size from 32k to 1 million tokens. Results are averaged over all constituent datasets in each task.

### 4.3 Audio Retrieval

Audio retrieval baseline used PALM 2 DE from [15], a dual-encoder model trained to maximize the similarity between audio and their transcription, and has achieved previous state-of-the-art on the FLEURS datasets. At present, GPT-4o does not support audio input.

**Results** Gemini-1.5-Pro demonstrates comparable performance to PALM 2 DE across all 5 languages (Table 2). We notice that Gemini-1.5 Pro notably surpasses PALM 2 DE in Hindi; this advantage likely stems from variations in pre-training data. Figure 5 further confirms Gemini-1.5-Pro's robust performance across various context length, highlighting the current capabilities of LCLMs while also indicating the need for more challenging audio datasets.

### 4.4 RAG

We set up a retrieve-and-read RAG pipeline as the baseline, using Gecko [24] for top-40 document retrieval, followed by Gemini-1.5-Pro for generating the answering conditioned on the question and the top documents.

**Results** Table 2 demonstrates that Gemini-1.5-Pro, with the entire corpus in context, outperforms the RAG pipeline on multi-hop datasets (HotpotQA and MuSiQue). This is because long-context model can reason over multiple passages in the context window using Chain-of-Thoughts [47], a capability that RAG pipelines typically lack without sophisticated planning and iterative retrieval mechanisms.

However, a specialized retriever like Gecko excels at ranking all topically relevant passages from a corpus, enabling it to identify a comprehensive set of passages covering all answers. This proves particularly beneficial for multi-target datasets, such as QUEST and QAMPARI.

Interestingly, Figure 5 reveals that LCLMs also demonstrate superior RAG performance at 200k and 1M context lengths compared to the RAG pipeline, even though their retrieval performance on the corresponding retrieval datasets is inferior to Gecko.

| Dataset | 32k | 128k/200k/1M |
|---------|-----|--------------|
| HotPotQA | 0.60 (-0.30) | 0.31 (-0.41) |
| MuSiQue | 0.20 (-0.60) | 0.10 (-0.43) |
| NQ | 0.60 (-0.10) | 0.37 (-0.44) |

Table 3: **Gemini's closed-book performance on RAG** (32k = development, rest = test queries). Red indicates the performance difference compared to the CiC prompting.

**Closed-Book Ablations** To further probe capabilities, we conduct closed-book ablations on Gemini 1.5 Pro, removing the corpus to assess LCLM performance based solely on parametric knowledge [27, 30]. Table 3 presents the results, revealing that the closed-book performance significantly lags behind our long-context and traditional model. This underscores the tested models' effectiveness in leveraging external information from the corpus to enhance its reasoning capabilities.

### 4.5 SQL-Like Compositional Reasoning

SQL baseline uses a semantic parser to tranlsate the natural langauge input into SQL query, then excute the SQL query over the database. Specifically, we use DAIL-SQL [14], a state-of-the-art semantic parser that prompts an LLM. We adapt DAIL-SQL by replacing its LLM with Gemini 1.5 Pro and using the fixed set of few-shot examples.

| Task (Metric) | Dataset | Best Prompt | Generic Instruction | Query at Beginning | Alphanumeric IDs | Titles Only | Without ID Echo | Corpus in Each Few-shot | Without CoT |
|---|---|---|---|---|---|---|---|---|---|
| **Text Retrieval (Recall@1)** | ArguAna | 0.84 | 0.76 | 0.72 | 0.81 | - | 0.78 | 0.62 | 0.79 |
| | FIQA | 0.79 | 0.77 | 0.58 | 0.75 | - | 0.76 | 0.78 | 0.85 |
| | NQ | 1.00 | 0.98 | 0.98 | 0.99 | 0.91 | 1.00 | 1.00 | 1.00 |
| | SciFact | 0.88 | 0.88 | 0.81 | 0.90 | 0.84 | 0.87 | 0.78 | 0.90 |
| **Text Set (mRecall@k)** | MuSiQue | 0.49 | 0.44 | 0.19 | 0.44 | 0.10 | 0.36 | 0.35 | 0.43 |
| | QAMPARI | 0.61 | 0.61 | 0.49 | 0.54 | 0.09 | 0.49 | 0.35 | 0.43 |
| | QUEST | 0.28 | 0.28 | 0.22 | 0.30 | 0.05 | 0.27 | 0.22 | 0.30 |
| **RAG (Span EM)** | MuSiQue | 0.53 | 0.55 | 0.39 | 0.50 | 0.23 | 0.54 | 0.48 | 0.50 |
| | NQ | 0.81 | 0.78 | 0.73 | 0.80 | 0.40 | 0.80 | 0.81 | 0.81 |
| | QAMPARI | 0.39 | 0.30 | 0.30 | 0.33 | 0.08 | 0.26 | 0.30 | 0.25 |
| | QUEST | 0.28 | 0.31 | 0.16 | 0.25 | 0.02 | 0.24 | 0.26 | 0.29 |
| **Average[†]** | | 0.59 | 0.57 | 0.47 | 0.56 | 0.30 | 0.54 | 0.54 | 0.55 |
| **(Δ)** | | - | (-0.02) | (-0.12) | (-0.03) | (-0.29) | (-0.05) | (-0.05) | (-0.04) |

Table 4: Ablation results of Gemini-1.5-Pro on different tasks of LOFT at 128k context length. Starting from our best prompt format (used in the rest of the experiments), individual facets of the corpus, query, and instruction are ablated to surface their relative effect on quality. [†]: The average is computed without ArguAna and FIQA, as not all ablations apply to them (they do not contain titles).

**Results** Results in Table 2 show that LCLMs achieve non-trivial performance, though they are significantly behind the text-to-SQL baseline. This reveals substantial headrooms to enhance the compositional reasoning capabilities of LCLMs.

**Reasoning Analysis** To gain insights into the short-comings of LCLMs in complex compositional reasoning, we categorize queries based on the operators in the gold SQL queries and measure Gemini-1.5-Pro's performance for each operator. Figure 6 shows that averaging is the most difficult operation while counting is relatively easy. Moreover, we find that reasoning over equality is considerably easier than reasoning over inequality.

### 4.6 Many-Shot ICL

**Results** Table 2 compares accuracy for Gemini 1.5 Pro and GPT-4o on all ICL benchmarks. For BBH, we report the accuracy on 32k, which is the maximum context length available. Gemini 1.5 Pro outperforms GPT-4o on all benchmarks, except for BBH-tracking7 where Gemini performs surprisingly poorly.

**Scaling Many Shot ICL** Fig. 7 illustrates the impact of increasing the number of examples on performance. In LIB-dialog, accuracy improves monotonically with more examples. In contrast, results on BBH are mixed. Knowledge-intensive tasks like BBH-date and BBH-salient see monotonic improvements similar to LIB-dialog, while reasoning-intensive tasks like BBH-tracking7 and BBH-web do not benefit. These results suggests that building and updating mental models is harder to learn from scaling the number of in-context examples.

## 5 CiC Prompt Ablations

We conduct ablations over the different facets of the CiC Prompt with ablated prompt examples in Appendix D. For the ablations, we evaluate Gemini-1.5-Pro at 128k context length.

The ablations show the effectiveness of our CiC prompting design. Removing tasks-specific instructions (`Generic Instruction`) or Chain-of-Thoughts reasoning (`Without CoT`) both lead to worse performance. We also observe performance decrease for `Corpus in Each Few-Shot`, where a small corpus (10 oracle and randomly passage) is added for each few shot example instead of using one shared corpus. Placing the query at the beginning of the prompt instead of the end (`Query at Beginning`) led to a significant and consistent performance decrease. This allows us to perform prefix-caching as we do not need to encode the corpus conditioned on the specific query.

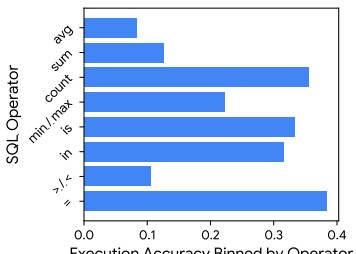

Figure 6: **SQL Reasoning Analysis**. We bin Spider queries by operators in their SQL query and report binned Gemini performance.

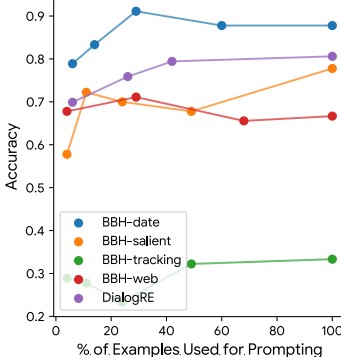

Figure 7: **ICL Performance** as we scale the percentage of examples used up to 100%.

For the document ID formatting, replacing monotonic numerical IDs with random (`Alphanumeric IDs`) negatively impacted performance in most datasets, possibly due to tokenizer being optimized for numerical values. Not repeating the ID at the end of the document (`Without ID Echo`) resulted in a 5% performance drop, confirming [39] that repeating text can compensate for missing context in autoregressive language models.

To test if model uses parametric knowledge instead of grounding on the context, we remove the document content and simply keep the document title and ID in the corpus (`Title Only`). Across all experiments, this ablation significantly degraded performance, indicating the model indeed relies provided context.

Finally, we study how the number of few-shot examples in the prompt affect quality in Figure 8. Increasing the number of examples increase quality overall on the retrieval task, from 0.76 at zero-shot to 0.81 at 5-shots.

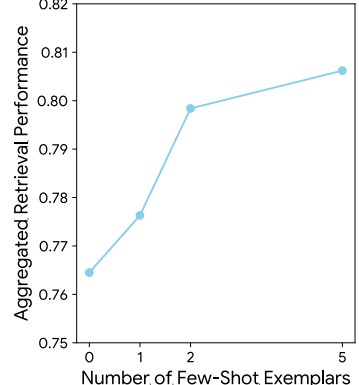

Figure 8: **Effect of the number of few-shot examples**. The performance increases with the number of few-shot examples.

## 6 Related Work

Evaluating long-context language models (LCLMs) remains a challenge due to the limitations of existing benchmarks. Many popular datasets and methods rely on synthetic tasks [41] such as the popular "Needle-in-A-Haystack" retrieval [19] or its extension to multi-hop QA [25]. While scalable to arbitrary lengths, these approaches do not fully capture the nuances of real-world retrieval or reasoning tasks [18]. Conversely, some recent benchmarks leverage existing NLP datasets for tasks such as extreme summarization and multi-document QA [6]. However, these lack the dynamic scaling capabilities of synthetic benchmarks.

LongAlpaca [10] and LongBench-Chat [5] evaluate instruction-following under long-text settings, while Ada-LEval [45] tests LCLMs on 100k+ tokens but with limited task diversity.

Closest to our work is [29], which applies LCLMs to long-context QA using top retrieved documents from MSMarco, similar to our RAG setup in LOFT. They find that LCLMs lose recall when relevant information is placed in the middle of the context (*i.e.,* lost-in-the-middle). However, their analysis is limited to contexts under 10k tokens. We extend the evaluation of LCLMs to up to 1M tokens context length and multiple modalities.

## 7 Conclusion

As language models improve and scale, their ability to retrieve and reason over increasing context lengths will unlock unprecedented use-cases. To measure this progress, we introduce LOFT, the Long Context Frontiers benchmark. LOFT is a suite of tasks that rigorously assesses LCLMs on tasks ripe for a paradigm shift: retrieval, retrieval-augmented generation, and SQL-like reasoning. LOFT provides dynamic scaling of context lengths, up to 1 million tokens, ensuring that evaluations remain relevant as LCLMs continue to evolve. Initial findings showcase that despite never trained to do retrieval, LCLMs have retrieval capabilities rivaling dedicated SOTA retrieval systems. Nevertheless, there remains considerable room for advancement in long-context reasoning, particularly as models gain access to even longer context windows. We believe that LOFT provides fertile testing ground for measuring progress in long-context modeling.

**Limitations** Our experiments were constrained by the speed, computational resources and financial costs associated with utilizing the long context language models. We were not able to measure the efficiency improvements from prefix caching [16] at the time of the experiments due to API constraints; without caching, Gemini-1.5-Pro API's median lantency is roughly 4 seconds on 32k token input, 12 seconds on 128k token input, and 100 seconds on 1m token input. Additionally, the scope of our retrieval and RAG tasks was limited to 1 million tokens, which still has a large gap towards real-world applications that may involve millions or even billions of documents.

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
