# A LOFT Dataset Creation

## A.1 Dataset Selection

**Text Retrieval & RAG**   We test single-document retrieval on a representative subset of the BEIR benchmark [47], prioritizing datasets with high-quality ground truth labels. We also include TopiOCQA [2], which is a multi-turn conversational retrieval dataset. We measure performance on single-document retrieval using Recall@1. Additionally, we test multi-document retrieval on HotPotQA [54], MuSiQue [48], QAMPARI [35], where a set of documents must be retrieved to answer the query. The evaluation metric for multi-document retrieval is mRecall@$k$, which gives a score of 1.0 if all $k$ gold set items are retrieved in top-$k$ and 0.0 otherwise. When creating the LOFT version of the multi-document retrieval datasets, we limit the number of relevant documents per query to $k = 2, 5, 5$, and 3 for HotPotQA, MuSiQue, QAMPARI, and QUEST, respectively, and the corresponding $k$'s are used for mRecall@$k$ (e.g. HotPotQA uses mRecall@2).

Our RAG task contains subsets of retrieval datasets, which have phrase-level answer annotations: Natural Questions, TopiOCQA, HotPotQA, MuSiQue, QAMPARI, and QUEST. We use subspan exact match (EM) [1] for evaluating performance of all the datasets. In case of multi-answer datasets (i.e. QAMPARI, QUEST), we first match predicted answers to gold standard answers based on whether they overlap [13] via linear sum assignment algorithm. We then give full credit if every gold answer has a perfect match with aligned predicted answers.

**Visual Retrieval**   We employ four diverse visual benchmarks: Flickr30k [55] and MSCOCO [31] for text-to-image retrieval; MSR-VTT [53] for text-to-video retrieval (sampling 3 frames per video); and OVEN [20] using the entity split for image-text retrieval where both queries and retrieval targets consist of image-text pairs. All images are resized to 512x512 and performance is assessed using Recall@1 for all datasets.

**Audio Retrieval**   We utilize a subset of the multilingual FLEURS dataset [12], focusing on the five most spoken languages[6]: English (en), Hindi (hi), Chinese (zh), Spanish (es), and French (fr). Recall@1 is employed as the evaluation metric, given the single gold target.

**SQL**   We evaluate SQL-like reasoning on Spider, a single-turn text-to-SQL dataset [58], and SparC, its multi-turn variant [59]. The input contains the database tables serialized as CSV and the natural language question. The model is allowed to perform reasoning in natural language before giving the final answer, which must be formatted in a Markdown code block. The extracted answers are evaluated against the execution results of the gold SQL queries. For SparC, the multi-turn questions are provided one-by-one in a conversational format, and credit is awarded only when the answers of all steps are correct.

**Many-shot ICL**   We investigate LCLMs' many-shot ICL capabilities by repurposing datasets from Big Bench Hard (BBH) [43, 44] and LongICLBench (LIB) [57, 30] to fit a many-shot ICL setting, focusing on multi-class classification tasks. The first set of datasets is drawn from Big-Bench Hard and includes: `date_understanding` (BBH-date), `salient_error_translation_detection` (BBH-salient), `tracking_shuffled_objects_seven_objects` (BBH-tracking7), and `web_of_lies` (BBH-web), each with up to 150 examples for prompting and up to 7 classes. Unlike other LOFT tasks, the full corpus fits within 32k tokens which leads us to also create variants from 2k to 32k context lengths. We use accuracy as our metric for Big Bench Hard. We also evaluate with DialogRE [57], a dialogue-based relation classification dataset with 36 relation labels. We follow the LongICLBench format but use accuracy as our metric.

---

[6]https://en.wikipedia.org/wiki/List_of_languages_by_total_number_of_speakers

# B   Datasets Processing Details

**Content Filtering**   The language model APIs often block inputs with potentially harmful contents. When creating LOFT, we tried to remove such contents from textual and visual inputs. Our filtering was done using a classifier as well as a keyword-based filtering. Despite our best effort, some API calls still refused to provide answers, which we treated as incorrect in our evaluation.

**Tokenization**   To measure the size of a corpus, we count the number of tokens returned by the SentencePiece tokenizer [24].

**Links to Dataset Sources**   LOFT repurposes existing datasets for evaluating LCLMs. Here are the links to the original datasets used in LOFT.

- Text Retrieval - BEIR (ArguAna, FEVER, FIQA, MS MARCO, NQ, Quora, SciFact, Touché-2020, HotPotQA) [47]: `https://github.com/beir-cellar/beir`
- Text Retrieval - TopiOCQA [2]: `https://github.com/McGill-NLP/topiocqa`
- Text Retrieval - MuSiQue [48]: `https://allenai.org/data/musique`
- Text Retrieval - QAMPARI [3]: `https://github.com/samsam3232/qampari`
- Text Retrieval - QUEST [35]: `https://github.com/google-research/language/tree/master/language/quest`
- Visual Retrieval - Flickr30k [55]: `https://www.kaggle.com/datasets/hsankesara/flickr-image-dataset`
- Visual Retrieval - MS COCO [31]: `https://cocodataset.org`
- Visual Retrieval - OVEN [20]: `https://github.com/open-vision-language/oven`
- Visual Retrieval - MSR-VTT [53]: `https://cove.thecvf.com/datasets/839`
- Audio Retrieval - FLEURS [12]: `https://huggingface.co/datasets/google/fleurs`
- RAG - Same as Text Retrieval
- SQL - Spider [58]: `https://yale-lily.github.io/spider`
- SQL - SparC [59]: `https://yale-lily.github.io/sparc`
- Many-Shot ICL - Big-Bench Hard [43, 44]: `https://github.com/suzgunmirac/BIG-Bench-Hard`
- Many-Shot ICL - LongICLBench [57, 30]: `https://github.com/TIGER-AI-Lab/LongICLBench`

 # C   Detailed Statistics

In Table 5, we show detailed statistics of the LOFT benchmark.

| Task | Dataset | # Queries (Few-shot / Development / Test) | Supported Context Length |
|---|---|---|---|
| **Text Retrieval** | ArguAna | 5 / 10 / 100 | 32k / 128k / 1M |
| | FEVER | 5 / 10 / 100 | 32k / 128k / 1M |
| | FIQA | 5 / 10 / 100 | 32k / 128k / 1M |
| | MS MARCO | 5 / 10 / 100 | 32k / 128k / 1M |
| | NQ | 5 / 10 / 100 | 32k / 128k / 1M |
| | Quora | 5 / 10 / 100 | 32k / 128k / 1M |
| | SciFact | 5 / 10 / 100 | 32k / 128k / 1M |
| | Touché-2020 | 5 / 10 / 34 | 32k / 128k / 1M |
| | TopiOCQA | 5 / 10 / 100 | 32k / 128k / 1M |
| | HotPotQA | 5 / 10 / 100 | 32k / 128k / 1M |
| | MuSiQue | 5 / 10 / 100 | 32k / 128k / 1M |
| | QAMPARI | 5 / 10 / 100 | 32k / 128k / 1M |
| | QUEST | 5 / 10 / 100 | 32k / 128k / 1M |
| **Visual Retrieval** | Flickr30k | 5 / 10 / 100 | 32k / 128k |
| | MS COCO | 5 / 10 / 100 | 32k / 128k / 1M |
| | OVEN | 5 / 10 / 100 | 32k / 128k / 1M |
| | MSR-VTT | 5 / 10 / 100 | 32k / 128k / 1M |
| **Audio Retrieval** | FLEURS-en | 5 / 10 / 100 | 32k / 128k |
| | FLEURS-es | 5 / 10 / 100 | 32k / 128k |
| | FLEURS-fr | 5 / 10 / 100 | 32k / 128k |
| | FLEURS-hi | 5 / 10 / 100 | 32k / 128k |
| | FLEURS-zh | 5 / 10 / 100 | 32k / 128k |
| **RAG** | NQ | 5 / 10 / 100 | 32k / 128k / 1M |
| | TopiOCQA | 5 / 10 / 100 | 32k / 128k / 1M |
| | HotPotQA | 5 / 10 / 100 | 32k / 128k / 1M |
| | MuSiQue | 5 / 10 / 100 | 32k / 128k / 1M |
| | QAMPARI | 5 / 10 / 100 | 32k / 128k / 1M |
| | QUEST | 5 / 10 / 100 | 32k / 128k / 1M |
| **SQL** | Spider | 1 / 10 / 100 | 32k / 128k / 1M |
| | SParC | 1 / 10 / 100 | 32k / 128k / 1M |
| **Many-Shot ICL** | BBH-date | - / 10 / 90 | 32k |
| | BBH-salient | - / 10 / 90 | 32k |
| | BBH-tracking7 | - / 10 / 90 | 32k |
| | BBH-web | - / 10 / 90 | 32k |
| | LIB-dialogue | - / 10 / 100 | 32k / 128k / 1M |

Table 5:   Tasks and datasets in the LOFT benchmark. We show the number of queries per each split and and supported context lengths for each dataset.

583

 # D  Dataset Instructions

| Dataset | Instruction |
|---------|-------------|
| **Text Retrieval** | |
| ArguAna | You will be given a list of statements. You need to read carefully and understand all of them. Then you will be given a claim, and your goal is to find all statements from the list that can counterargue the claim. |
| FEVER Scifact | You will be given a list of passages. You need to read carefully and understand all of them. Then you will be given a claim, and your goal is to find all passages from the list that can help verify the claim as true of false. |
| FIQA MS MARCO NQ, TopiOCQA | You will be given a list of documents. You need to read carefully and understand all of them. Then you will be given a query, and your goal is to find all documents from the list that can help answer the query. |
| Quora | You will be given a list of questions. You need to read carefully and understand all of them. Then you will be given a new question, and your goal is to find all questions from the list that are near duplicates of the new question. |
| Touché-2020 | You will be given a list of arguments. You need to read carefully and understand all of them. Then you will be given a controversial debating topic, and your goal is to find arguments from the list that's relevant to the topic. |
| HotPotQA MuSiQue QAMPARI QUEST | You will be given a list of documents. You need to read carefully and understand all of them. Then you will be given a query that may require you to use 1 or more documents to find the answer. Your goal is to find all documents from the list that can help answer the query. |
| **Visual Retrieval** | |
| Flickr30k MS COCO | You will be given a list of images. You need to carefully watch all of them. Then you will be given a new sentence, and your goal is to find most relevant image from the list for the given sentence. |
| OVEN | You will be given a list of Wikipedia entries which contains Wikipedia ID, Title and Description image. You need to carefully watch all of them. Then you will be given a input image and a question related to the image, and your goal is to find most relevant Wikipedia entry from the list that can be used to best answer the question. |
| MSR-VTT | You will be given a list of videos which contains the video ID and video content (present as sequence of images, with timestamp in text). You need to carefully watch all of them. Then you will be given a text query, and your goal is to find most relevant video from the list that can best answer the question. |
| **Audio Retrieval** | |
| FLEURS-* | You will be given a list of audio which contains Audio ID and audio. You need to carefully listen all of them. Then you will be given a transcript, and your goal is to find most relevant audio from the list that matches the given transcript. Print out the Audio ID of the audio presented in the list. |
| **SQL** | |
| Spider SparC | You will be given a list of tables. You need to read all of the rows of each table. Then you will be given a query, and your goal is to get the answer from the tables. Then format the answer into a list of lists. When formatting the answer into a list of lists, make sure you use the exact fields that are provided in the tables. |

Table 6:  Instructions used for each LOFT dataset. We omit instructions for the RAG datasets, which are almost identical to text retrieval instructions. The ICL task does not use additional instructions, but only many-shot examples in their context.

 # E  Positional Analysis Detailed Results

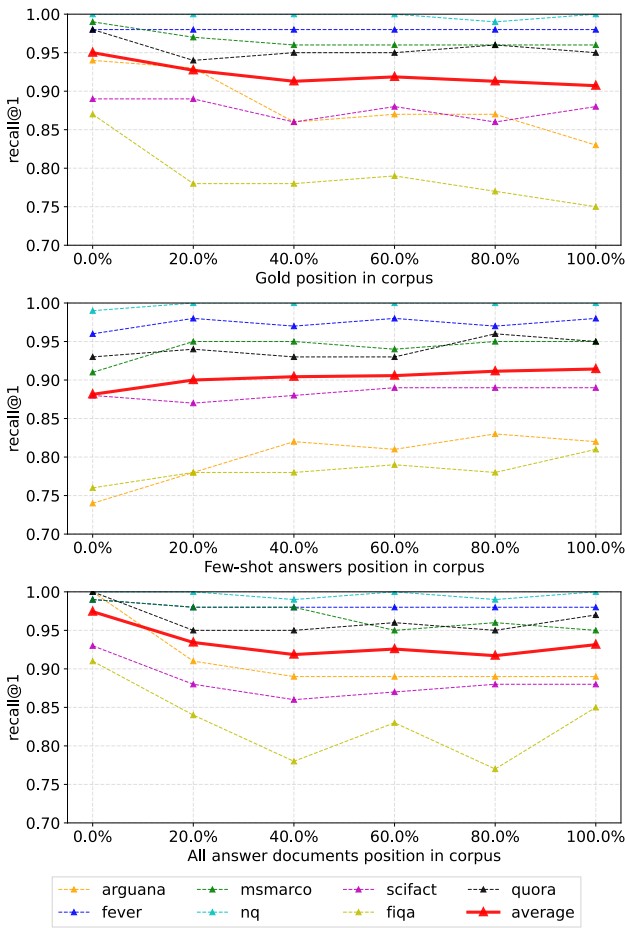

Figure 9: Detailed metrics of the positional analysis, where we vary the position of gold documents (needed for the answer) and few-shot documents (used in few-shot demonstrations). ***Top:*** we vary the gold documents position within the corpus. ***Middle:*** we vary the few-shot documents position within the corpus. ***Bottom:*** we group the gold and few-shot documents together, and vary their position within the corpus. The average is shown in red.

## F   Ablated Prompt Examples

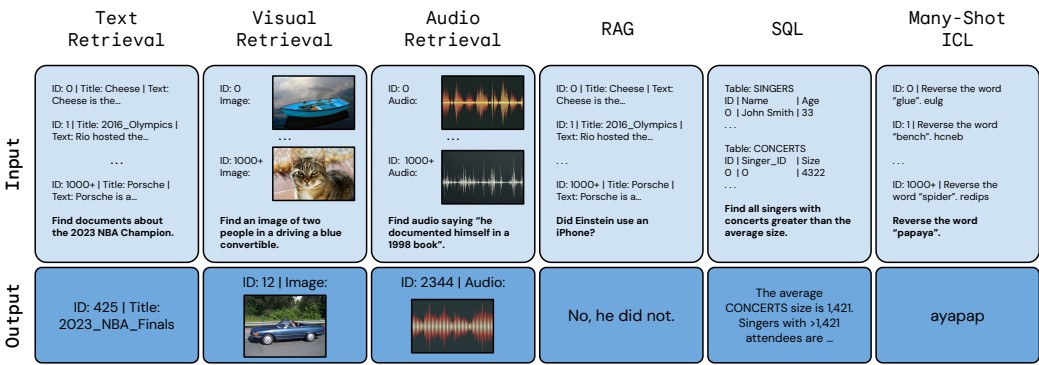

Figure 10:   Examples of the task prompts in LOFT. Each LCLM is expected to do in-context retrieval, reasoning, and many-shot learning on corpora up to millions of tokens.

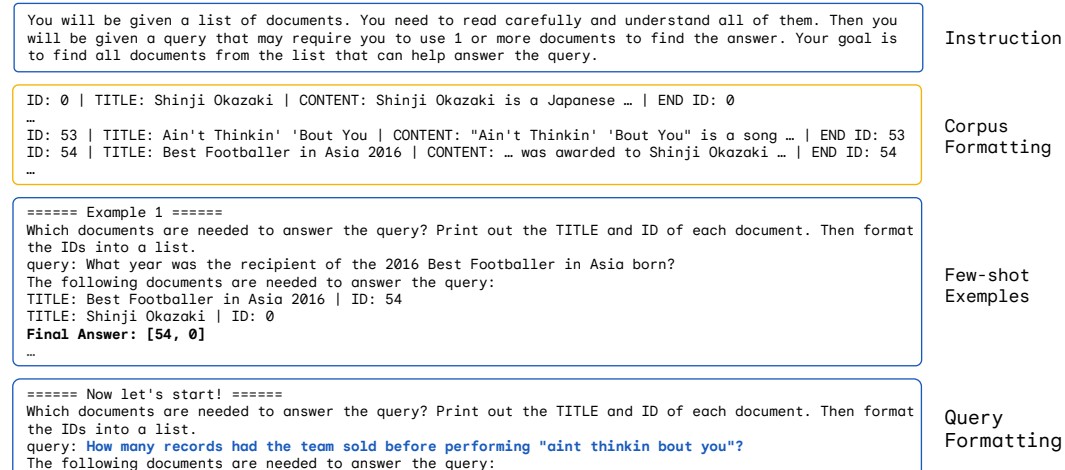

Figure 11: Original CiC prompt for HotPotQA, a retrieval dataset in LOFT. The prompt contains an instruction, a corpus, few-shot examples and a query.

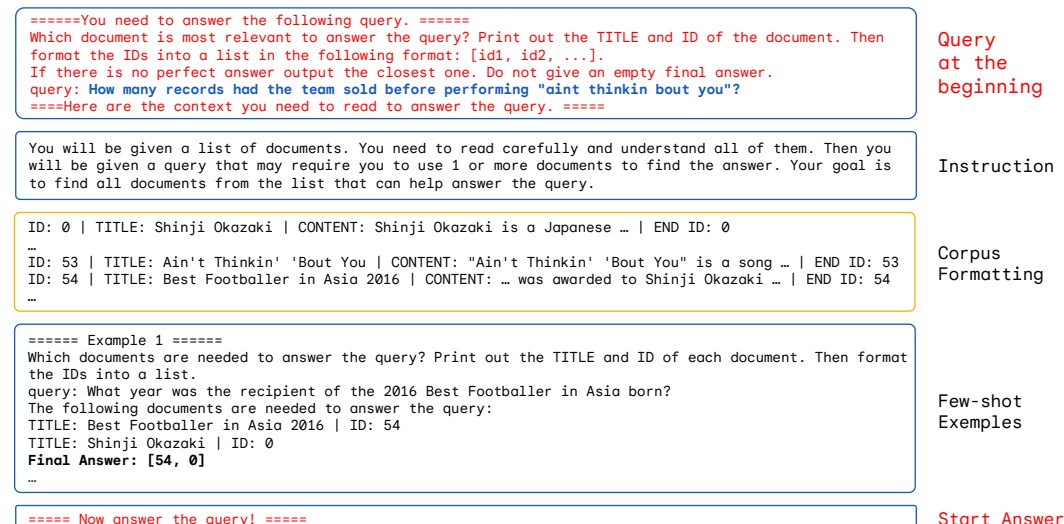

Figure 12: Generic Instruction Ablation, with changes to the original CiC prompt in red. The instruction is changed to a generic one that applies to all tasks in LOFT.

Figure 13: Query at the Beginning Ablation, with changes to the original CiC prompt in red. The query is placed at the beginning instead of the end.

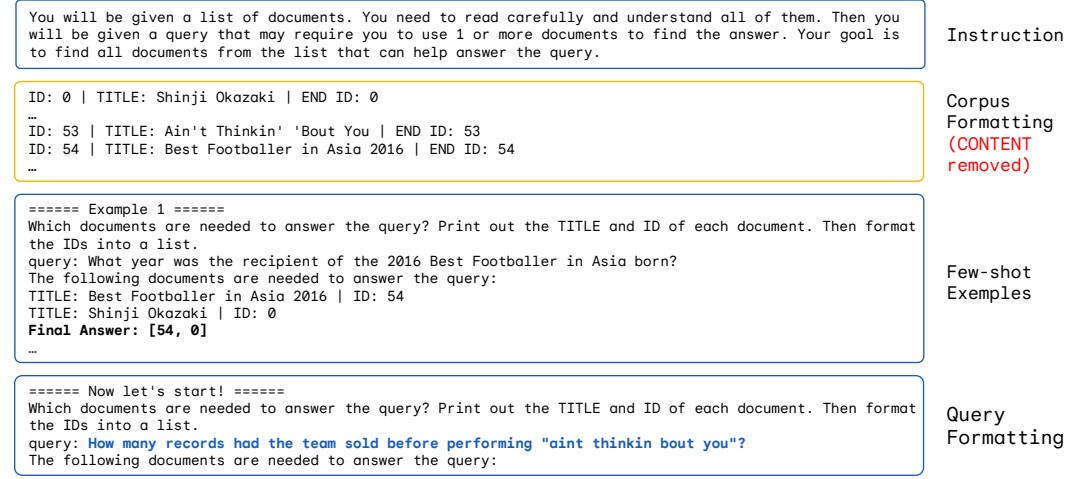

```
You will be given a list of documents. You need to read carefully and understand all of them. Then you will
be given a query that may require you to use 1 or more documents to find the answer. Your goal is to find
all documents from the list that can help answer the query.
```
Instruction

```
ID: D5Y5 | TITLE: Shinji Okazaki | CONTENT: Shinji Okazaki is a Japanese … | END ID: D5Y5
…
ID: y2h8 | TITLE: Ain't Thinkin' 'Bout You | CONTENT: "Ain't Thinkin' 'Bout You" is a song … | END ID: y2h8
ID: E8J2 | TITLE: Best Footballer in Asia 2016 | CONTENT: … was awarded to Shinji Okazaki … | END ID: E8J2
…
```
Corpus Formatting

```
====== Example 1 ======
Which documents are needed to answer the query? Print out the TITLE and ID of each document. Then format
the IDs into a list.
query: What year was the recipient of the 2016 Best Footballer in Asia born?
The following documents are needed to answer the query:
TITLE: Best Footballer in Asia 2016 | ID: E8J2
TITLE: Shinji Okazaki | ID: D5Y5
Final Answer: [E8J2, D5Y5]
…
```
Few-shot Exemples

```
====== Now let's start! ======
Which documents are needed to answer the query? Print out the TITLE and ID of each document. Then format
the IDs into a list.
query: How many records had the team sold before performing "aint thinkin bout you"?
The following documents are needed to answer the query:
```
Query Formatting

Figure 14: Alphanumeric Document ID Ablation, with changes to the original CiC prompt in red. Instead of using sequential numeric document IDs, a unique random alphanumeric ID is generated with alternating ASCII letters and digits.

```
You will be given a list of documents. You need to read carefully and understand all of them. Then you
will be given a query that may require you to use 1 or more documents to find the answer. Your goal is
to find all documents from the list that can help answer the query.
```
Instruction

```
ID: 0 | TITLE: Shinji Okazaki | END ID: 0
…
ID: 53 | TITLE: Ain't Thinkin' 'Bout You | END ID: 53
ID: 54 | TITLE: Best Footballer in Asia 2016 | END ID: 54
…
```
Corpus Formatting (CONTENT removed)

```
====== Example 1 ======
Which documents are needed to answer the query? Print out the TITLE and ID of each document. Then format
the IDs into a list.
query: What year was the recipient of the 2016 Best Footballer in Asia born?
The following documents are needed to answer the query:
TITLE: Best Footballer in Asia 2016 | ID: 54
TITLE: Shinji Okazaki | ID: 0
Final Answer: [54, 0]
…
```
Few-shot Exemples

```
====== Now let's start! ======
Which documents are needed to answer the query? Print out the TITLE and ID of each document. Then format
the IDs into a list.
query: How many records had the team sold before performing "aint thinkin bout you"?
The following documents are needed to answer the query:
```
Query Formatting

Figure 15: Title Only Ablation, with changes to the original CiC prompt in red. In this ablation, the document content is removed, keeping only the document title.

```
You will be given a list of documents. You need to read carefully and understand all of them. Then you
will be given a query that may require you to use 1 or more documents to find the answer. Your goal is
to find all documents from the list that can help answer the query.
```
Instruction

```
ID: 0 | TITLE: Shinji Okazaki | CONTENT: Shinji Okazaki is a Japanese …
…
ID: 53 | TITLE: Ain't Thinkin' 'Bout You | CONTENT: "Ain't Thinkin' 'Bout You" is a song …
ID: 54 | TITLE: Best Footballer in Asia 2016 | CONTENT: … was awarded to Shinji Okazaki …
…
```
Corpus
Formatting
(END ID
removed)

```
====== Example 1 ======
Which documents are needed to answer the query? Print out the TITLE and ID of each document. Then format
the IDs into a list.
query: What year was the recipient of the 2016 Best Footballer in Asia born?
The following documents are needed to answer the query:
TITLE: Best Footballer in Asia 2016 | ID: 54
TITLE: Shinji Okazaki | ID: 0
Final Answer: [54, 0]
…
```
Few-shot
Exemples

```
====== Now let's start! ======
Which documents are needed to answer the query? Print out the TITLE and ID of each document. Then format
the IDs into a list.
query: How many records had the team sold before performing "aint thinkin bout you"?
The following documents are needed to answer the query:
```
Query
Formatting

Figure 16: ID Echo Ablation, with changes to the original CiC prompt in red. In this ablation, the ID is only mentioned at the beginning of each document, and we remove the ID echo at the end (e.g. "END ID:").

```
You will be given a list of documents. You need to read carefully and understand all of them. Then you
will be given a query that may require you to use 1 or more documents to find the answer. Your goal is
to find all documents from the list that can help answer the query.
```
General
Instruction

```
====== Example 1 ======
ID: 0 | TITLE: Best Footballer in Asia 2016 | CONTENT: … was awarded to Shinji Okazaki … | END ID: 0
ID: 1 | TITLE: Shinji Okazaki | CONTENT: Shinji Okazaki is a Japanese … | END ID: 1
…
ID: 9 | TITLE: The Lodger (2009 film) | CONTENT: The Lodger is … | END ID: 9
Which documents are needed to answer the query? Print out the TITLE and ID of each document. Then
format the IDs into a list.
query: What year was the recipient of the 2016 Best Footballer in Asia born?
The following documents are needed to answer the query:
TITLE: Best Footballer in Asia 2016 | ID: 0
TITLE: Shinji Okazaki | ID: 1
Final Answer: [0, 1]
…
```
Few-shot
Exemples
With Sampled
Corpus

```
ID: 0 | TITLE: Shinji Okazaki | CONTENT: Shinji Okazaki is a Japanese … | END ID: 0
…
ID: 53 | TITLE: Ain't Thinkin' 'Bout You | CONTENT: "Ain't Thinkin' 'Bout You" is a song … | END ID: 53
ID: 54 | TITLE: Best Footballer in Asia 2016 | CONTENT: … was awarded to Shinji Okazaki … | END ID: 54
…
```
Corpus
Formatting

```
====== Now let's start! ======
Which documents are needed to answer the query? Print out the TITLE and ID of each document. Then format
the IDs into a list.
query: How many records had the team sold before performing "aint thinkin bout you"?
The following documents are needed to answer the query:
```
Query
Formatting

Figure 17: Corpus in Each Few-shot Ablation, with changes to the original CiC prompt in red. In particular, in this ablation, each few-shot example contains a sampled corpus (10 documents), the full corpus is then given before the Query part of the prompt.

Figure 18: No Chain-of-Thought Ablation, with changes to the original CiC prompt in red. In particular, in this ablation, we remove the chain-of-thought from the few-shot examples, instead opting to output the answer directly.

 # G   Qualitative Analysis of Long-Context Wins and Losses

| Task = **Retrieval** | |
| --- | --- |
| Dataset = **Natural Questions** 1M | |
| **Input** **Output** | `CiC-prompt`(query="when does monday night raw come on hulu")
The following documents can help answer the query:
TITLE: WWE Raw \| ID: 1023
Final Answer: ['1023'] (✓) |
| **Input** **Output** | `CiC-prompt`(query="where does the synthesis of new dna from existing dna occurs")
The following documents can help answer the query:
TITLE: DNA replication \| ID: 3355
Final Answer: ['3355'] (✓) |
| **Input** **Output** | `CiC-prompt`(query="jawaharlal nehru centre for advanced scientific research jakkur campus")
TITLE: Jawaharlal Nehru Centre for Advanced Scientific Research \| ID: 6911
Final Answer: ['6911'] (✗) // Incorrect ID (gold = 5691) |
| Dataset = **HotPotQA** 128k (multi-hop) | |
| **Input** **Output** | `CiC-prompt`(query="Which internationl football team has had a player endorse the Nike Hypervenom football boot?")
The following documents can help answer the query:
reasoning: The Nike Hypervenom is endorsed by players such as Marcus Rashford based on the document **ID:** 413 (title: Nike Hypervenom).
reasoning: Marcus Rashford plays for the England national team based on the document **ID:** 558 (title: Marcus Rashford).
Final Answer: ['413', '558'] (✓) |
| **Input** **Output** | `CiC-prompt`(query="What NHL team drafted Kaspars Astasenko in 1999 and won the Stanley Cup in 2003-04?")
The following documents can help answer the query:
ID: 217 \| TITLE: Kaspars Astašenko \| CONTENT: Kaspars Astašenko (17 February 1975 – 20 November 2012) was a Latvian professional ice hockey player. Astašenko was born in Riga, Latvia.  Astašenko was drafted by the Tampa Bay Lightning in the 1999 NHL Entry Draft, 127th overall. Astašenko played parts of two seasons in the National Hockey League with the Lightning. \| END ID: 217
ID: 508 \| TITLE: Tampa Bay Lightning \| CONTENT: The Tampa Bay Lightning are a professional ice hockey team based in Tampa, Florida. It is a member of the Atlantic Division of the Eastern Conference of the National Hockey League (NHL). The Lightning have one Stanley Cup championship in their history, in 2003–04. The team is often referred to as the Bolts, and the nickname is used on the current third jersey. The Lightning plays home games in the Amalie Arena in Tampa. \| END ID: 508
Final Answer: ['217', '508'] (✓) |
| **Input** **Output** | `CiC-prompt`(query="What drum machine was introduced in 1989 and allowed 32 notes to be played simultaneously?")
The following documents can help answer the query:
ID: 773 \| TITLE: Roland R-8 \| CONTENT: The R-8 Human Rhythm Composer is an electronic drum machine introduced in 1989 by Roland Corporation, using PCM voices. ... The device has eight individual outputs, 32-voice polyphony, and four-part multitimbral MIDI. \| END ID: 773
Final Answer: ['773'] (✗) // Partially correct (666 is missing) |

Table 7: Qualitative examples showcasing the successes and failures of the CiC prompting in the text retrieval task. We use Gemini-1.5 Pro for the analysis. For HotPotQA, we observed that some of the questions are actually answerable from a single document as illustrated in the last example.

| | Task = **RAG** |
|---|---|
| | **Dataset = MuSiQue** 128k (multi-hop) |
| **Input** | `CiC-prompt`(query="Where do greyhound buses leave from in the city the performer of Born A Lion formed in?") |
| **Output** | reasoning: Born A Lion is the debut album by Canadian rock band Danko Jones. |
| | reasoning: Danko Jones is a Canadian rock trio from Toronto, Ontario. |
| | reasoning: The Toronto Coach Terminal is the central bus station for inter-city services in Toronto, Ontario, Canada. |
| | ID: 427 \| TITLE: Born a Lion |
| | ID: 383 \| TITLE: Danko Jones |
| | ID: 724 \| TITLE: Toronto Coach Terminal |
| | Final Answer: ['Toronto Coach Terminal'] (✓) |
| **Input** | `CiC-prompt`(query="When did the rx 350 model of the luxury division of the company that built Daihatsu boon change body style?") |
| **Output** | reasoning: The Daihatsu Boon is also sold as the Toyota Passo. |
| | reasoning: The Passo is sold at "Toyota Corolla Store" Japanese dealerships. |
| | reasoning: Scion is a discontinued marque of Toyota. |
| | reasoning: The Lexus RX is a mid-size luxury crossover SUV sold since 1998 by Lexus, a luxury division of Toyota. |
| | reasoning: The first generation Lexus RX was introduced in 1998 and was produced until 2003. |
| | reasoning: The second generation Lexus RX was introduced in 2003 and was produced until 2009. |
| | reasoning: The third generation Lexus RX was introduced in 2008 and was produced until 2015. |
| | reasoning: The fourth generation Lexus RX was introduced in 2015. |
| | reasoning: The RX 350 model was introduced in the second generation Lexus RX in 2003. |
| | Final Answer: ['2003', '2009', '2015'] (✗) // Incorrect (answer = 'Sales began worldwide in April 2012') |
| | **Dataset = QUEST** 128k (multi-answer) |
| **Input** | `CiC-prompt`(query="History books about Malaysia or the Qing dynasty or about South Korea") |
| **Output** | reasoning: Chronicle of Malaysia gives an account of Malaysia from January 1957 to 31 August 2007. |
| | reasoning: Treason by the Book is a historical account of the Zeng Jing case which took place during the reign of the Yongzheng Emperor of Qing China around 1730. |
| | reasoning: Korea: A Walk Through the Land of Miracles is a book by Simon Winchester. He recounts his experience walking across South Korea, from Jeju in the south to the DMZ in the north. |
| | Final Answer: ['Chronicle of Malaysia', 'Korea: A Walk Through the Land of Miracles', 'Treason by the Book'] (✓) |
| **Input** | `CiC-prompt`(query="Films about sisters shot in Paris") |
| **Output** | reasoning: Peppermint Soda is a 1977 French comedy-drama film directed by Diane Kurys. The film follows two teenage sisters over the course of the year 1963. The high school where the film takes place is the Lycée Jules-Ferry in Paris, France. |
| | Final Answer: ['Peppermint Soda'] (✗) // Partially correct (missing 'Le Divorce'and 'Two English Girls') |

Table 8: Qualitative examples showcasing the successes and failures of the CiC prompting in the RAG task. We use Gemini-1.5 Pro for the analysis. In each example, we underline correct reasoning chains provided by the model.

| Task = **SQL** |
| --- |

| Dataset = **Spider** 1M |
| --- |

| **Input** | CiC-prompt(query="Find the year which offers the largest number of courses.") |
| --- | --- |
| **Output** | Here's a step-by-step approach using the provided tables: |

**1. Identify relevant data:**
We need to look at the "year" column in the "section" table, which lists the year each course section was offered.

**2. Count courses per year:**
Group the data by "year" and count the number of unique "course_id" values for each year. This will give us the number of distinct courses offered in each year.

**3. Find the year with the maximum count:**
Identify the year with the highest count of distinct courses.

**Based on the data provided, here's the breakdown of courses per year:**

| Year | Number of Courses |
|—|—|
| 2001 | 29 |
| 2002 | 36 |
| 2003 | 38 |
| 2004 | 31 |
| 2005 | 31 |
| 2006 | 41 |
| 2007 | 40 |
| 2008 | 37 |
| 2009 | 35 |
| 2010 | 39 |
**The year 2006 has the largest number of courses offered (41).**

Final Answer: ['2006'] (✔)

| **Input** | CiC-prompt(query="List total amount of invoice from Chicago, IL.") |
| --- | --- |
| **Output** | Here's a step-by-step approach using the provided tables: |

**1. Identify relevant data:**
* **Invoices:** This table contains the billing city and total for each invoice.

**2. Filter invoices from Chicago, IL:**
* Scan through the "billing_city" column and identify invoices where the city is "Chicago" and the state is "IL".
* **Based on the data provided, invoices 24, 92, 103, 158, 287, 310, 332, and 384 are from Chicago, IL.**

**3. Sum the total amounts:**
* Add up the "total" values for the invoices identified in step 2.
* **The total amounts for the invoices are 1.98, 1.98, 15.86, 8.91, 1.98, 7.96, 5.94, and 0.99, respectively.**
* **The sum of these amounts is 36.6.**

**4. Present the result:**
* The final output will be a single value representing the total amount of invoices from Chicago, IL.

Final Answer: ['36.6'] (✗)

Table 9: Qualitative examples showcasing the successes and failures of the CiC prompting in the SQL task. We use Gemini-1.5 Pro for the analysis. In SQL, the long-context model first outputs a reasoning chain in natural language ideally simulating the execution of the SQL query before arriving at the prediction.