# OpenReview forum: "Can Long-Context Language Models Subsume Retrieval, SQL, and More?"
_NeurIPS.cc/2024/Datasets_and_Benchmarks_Track — Submitted to NeurIPS 2024 Track Datasets and Benchmarks_

### Official Review · Reviewer_LA2k · 2024-07-18
**LOFT - A benchmark for Long Context Language Models**

**Rating:** 6
**Confidence:** 4
**Clarity:** The paper is well written with no lap…

**Review:**

Quality:  The paper was of good quality, had valid examples, comparisons, and plausible explanations of model behavior. The diverse set of 6 tasks chosen for this benchmark covered most available use cases for LCLMs, and were each individually documented well.

Clarity: The language was crisp and clear. Figures, Tables, and Appendices were formatted well and their titles conveyed the exact purpose of the illustration. The paper signifies advantages and disadvantages quite clearly as well.

Originality: The originality of this benchmark came through their introduced CiC prompting strategy. They conducted a detailed ablation study for the prompting method.

The results for SQL like reasoning were not up to the mark of the traditional trained text to sql models. They not only observed this but also have done an operator based analysis of where the queries fail, which was thorough. However, there could have been added illustrated examples of where and why the SQL queries fail.

One of my main concerns is if using long context is in general a sensible idea and especially so in the area of databases.
Should one really process an 'entire' database for every query (e.g., CIC prompting with the entire corpus)? Every token comes at a computational cost and it seems unreasonable to add so much content (which might often be the same) each time to answer a query.

**Strengths:**

- Showcase the strengths of LCLMs - Despite never being trained to do specific tasks, purely based out of large context windows, LCLMs outperform their specific baselines in many tasks.

- Detailed Ablation studies for Corpus in context prompting with variations in instruction methodologies (CoT reasoning, context specific instructions), and how they affect performance.

- Showcased clear limitations. Given today’s scale of data, it is not practical as of now to use LCLMs as opposed to traditional models for large corpus. Gemini / GPT APIs give out large latencies as well.

- All datasets are open source and the benchmark should be reproducible.

- Choice of tasks are comprehensive and diverse, hence proving useful to the research community.

- Futuristic vision

**Additional Feedback:**

None

**Correctness:**

The experimental setup was in line with the required results for the benchmark.
The task-specific evaluation metrics were taken based out of the task’s most used evaluation metric.

**Documentation:**

There is enough detail on data collection and organization. The authors plan to make the dataset and code public - which of course needs to happen to make this a benchmark paper.

**Limitations:**

Yes, limitations have been addressed.
No seen negative impact to the society or the research community.

The problems were mainly relating to lack of compute and LCLM API limitations.

**Opportunities For Improvement:**

As mentioned earlier, the actual performance of the generated SQL is not great (yet). And that is fine, given the title of the paper as well. The answer to the question is probably something along the lines "No, not yet, but there is an interesting potential."
To move this approach forward a more thorough error analysis would help understanding the challenges better.

Adding a longer discussion that it would make sense to consider approaches like RAG and Search (in the data discovery sense) to reduce the required context length drastically and therefore computational burden.

Not sure if this counts as an improvement, but did you consider using SPARQL instead of SQL? It might be that this 'richer' language might be beneficial in the LOFT setting as well?
E.g.: "Leveraging small language models for Text2SPARQL tasks to improve the resilience of AI assistance", https://arxiv.org/abs/2405.17076

**Relation To Prior Work:**

Yes, there is a comprehensive discussion of how their work differs from their closest related works.

**Summary And Contributions:**

- LOFT - A benchmark for Long Context Language Models containing a diverse set of tasks that require large context lengths (between 32k - 1mil tokens)

- 6 Tasks comprising of 35 datasets include : Text, video, audio, SQL, RAG, and In - context learning.
- Introduced a prompting standard CiC (Corpus in Context) to prompt long context models.

- All experiments done on the most recent LCLMs - Gemini 1.5 Pro and GPT-4o.

- Comparing specialized models vs LCLMs performance for all tasks.

- Positional analysis of CiC prompting has been showcased. Placement of query and few-shot examples matters.

- Comprehensive analysis of advantages and disadvantages of LCLMs - where they outperform traditional baselines, and where they fail.

---

> ### Author Rebuttal · Authors · 2024-08-17
>
> Thank you for your constructive comments. Please see our responses below for your concerns.
>
> 1. **One of my main concerns is if using long context is in general a sensible idea and especially so in the area of databases. Should one really process an 'entire' database for every query?**
>
> In general, probably not at the moment! A semantic parser would be faster and less error prone. Instead, we use the SQL task as an evaluation to stress-test LCLM’s reasoning and math capabilities. As found in our study, LCLMs still have a lot of headroom in these areas. This finding can guide LCLM developers to improve these general capabilities, which the SQL task measures.
>
> We agree with the reviewer that currently it is not practical to put an entire database into the prompt. However, looking into the future, this could become an interesting alternative as LCLMs get smarter and more efficient.  For example, using LCLMs for complex data science problems in the future might involve pipelines with many fewer steps if much of the data analysis can be done through simple prompting. It is important to start evaluating now and that’s why we create LOFT.
>
> Consider also some realistic cases downstream users might face, such as if they are given a database with some formatting errors but only needed to execute a single query against it. In that case, it might actually be easier to use an LCLM than trying to fix the database. There might also be other similar types of database formats that are less well studied than SQL, or a mixture of structured and unstructured data, or different data across modalities. We use existing SQL datasets as a proxy for the type of reasoning we expect will be required by users of LCLMs in the future.
>
> 2. **To move this approach forward a more thorough error analysis would help understanding the challenges better.” “there could have been added illustrated examples of where and why the SQL queries fail.**
>
> Thanks for the suggestion! We will include a more thorough analysis on the SQL task. We did provide two examples in the supplementary material, but we will add some additional examples.
>
> Here’s one example (from our supplementary material):
>
> query="List total amount of invoice from Chicago, IL."
>
> Output:
> ```
> Here’s a step-by-step approach using the provided tables:
> **1. Identify relevant data:**
> * **Invoices:** This table contains the billing city and total for each invoice.
>
> **2. Filter invoices from Chicago, IL:**
> * Scan through the “billing_city” column and identify invoices where the city is “Chicago” and the state is “IL”.
> * **Based on the data provided, invoices 24, 92, 103, 158, 287, 310, 332, and 384 are from Chicago, IL.**
>
> **3. Sum the total amounts:**
> * Add up the "total" values for the invoices identified in step 2.
> * **The total amounts for the invoices are 1.98, 1.98, 15.86, 8.91, 1.98, 7.96, 5.94, and 0.99, respectively.**
> * **The sum of these amounts is 36.6.** (authors’ comment: sum is incorrect)
>
> **4. Present the result:**
> * The final output will be a single value representing the total amount of invoices from Chicago, IL.
>
> Final Answer: ['36.6'] (authors’ comment: answer is wrong)
> ```
>
> 3. **Not sure if this counts as an improvement, but did you consider using SPARQL instead of SQL**
>
> This is a great suggestion! The benchmark looks indeed interesting and worth testing. As also suggested by other reviewers, we will expand our database-related tasks to make the evaluation more comprehensive.
>
> 4. **Adding a longer discussion that it would make sense to consider approaches like RAG and Search (in the data discovery sense) to reduce the required context length drastically and therefore computational burden.**
>
> I believe this question is asking about the tradeoff between putting an (entire) corpus in context vs. retrieving a subset of the corpus first and then placing the retrieved portion in context.
>
> We want to emphasize that the primary goal of the LOFT benchmark is to evaluate fundamental LCLM capabilities, by testing how well LCLMs can solve these tasks without using tools like RAG and search. As shown in our paper, LCLMs’ quality is already on par with RAG baselines without any additional development effort.
>
> That said, for practical downstream applications, it would likely make sense to combine with other tools in order to perform these tasks. We are happy to add a longer discussion about these alternative design choices and what the computational tradeoff would be.
>
> 5. **The authors plan to make the dataset and code public - which of course needs to happen to make this a benchmark paper.**
>
> While our dataset was initially omitted from the paper for anonymity purposes, we've found that full anonymization isn't necessary for this track. Therefore, we're staring it with the reviewers: https://github.com/google-deepmind/loft

---

> > ### Author Response · Authors · 2024-08-21
> >
> > We'd like to kindly request you to take a look at our rebuttal and provide your updated review. Thank you for providing valuable feedback on our paper.

---

### Official Review · Reviewer_VBfb · 2024-07-22
**Interesting problem**

**Rating:** 2
**Confidence:** 5

**Review:**

### Pros:

1. Authors collect and conclude many tasks from existing datasets to test LCMs performance.
2. Authors post an interesting research problem settings to evaluate LCLMs on IR-based problems.

### Cons

1. No dataset contributions. The benchmark is constructed by including all existing datasets. There are no efforts to create the data on their own. Some of the data is not very new, which leaves an obvious concern that conclusions and observations are not trustworthy or reliable since it's hard to tell how much data leakage problems matter. For example, Gemini-1.5-Pro can achieve 100% on NQ, which is very interesting. Any dataset and benchmark will include issues like ambiguity, and gold errors, so does 100% accuracy prove that Gemini-1.5-pro already memorizes most knowledge of this dataset? And if so how could authors make the results reliable to the public?
2. The motivation of this paper doesn't make sense to me. In the Abstract and Introduction, the authors stressed that the advantages of the long-context model LLM are that it can feed the entire corpus of information and eliminate the need for knowledge of tools. However, how such LLMs can deal with the temporal knowledge out-of-date problems? The motivation may be over claimed.
3. The numbers of the paper are not very clear. First, in Table 1, it's very abnormal to see that two text-to-SQL datasets can share the same length of the inputs. And in Table 2, it's very interesting to see LCLMs already achieved 100% and 92% in NQ. Also according to the leaderboard in MSMARCO, https://microsoft.github.io/MSMARCO-Document-Ranking-Submissions/leaderboard/, the SOTA performance only achieves 51, so it's still the risk that LCLMs already memorize such classical and old benchmarks.
4. SQL parts are totally problematic.
    - Just one data resource will lead to biased conclusions. It seems the authors involve two text-to-SQL datasets, but SparC actually is a multi-turn version of Spider, which only decomposes some hard questions into sub-questions or queries with the same dev databases. This means authors only involve one evaluation system and data resources, which are biased settings for other researchers to get fair conclusions. More recent datasets should be involved, such as KaggleDBQA, EHRSQL, BIRD, SEDE, etc. Compared to the IR and RAG part, only two datasets with the same databases from the same group of people are very biased settings.
    - Authors implemented a very unrealistic setting for SQLs. In order to fit the long-context setting, authors choose to feed all tables including values (entire) databases into LCLMs, which is quite unrealistic. First, all text-to-SQL papers in the era of LLMs will only provide DB schema including constraints and sampled values. Even the paper [3], proves that not always larger sample rows could lead to better performance. I don't know the motivation for this. I think it Sacrifices realism too much in order to simplify settings. Second, it's impossible to load all database values in realistic scenarios since authors steadily stress on 'realistic'. Databases of Spider and SparC only contain ~2k rows for each, while more databases contain more than 1 million rows. How to feed them all to your setting? Also in the Spider paper [4], they stated that their databases are collected from textbooks, schema only from from (http://www.databaseanswers.org/) and fill the fake value by (http://filldb.info/). Which resources of databases do authors think are realistic? However, some recent text-to-SQL benchmarks such as KaggleDBQA [6], EHRSQL [7], and BIRD-SQL [8] are collected from the realistic data resources from real applications or data science platforms are not included.
    - Evaluation Setting is questionable. The authors choose to use execution accuracy as a metric. However, GPT-4O only gets 14%, while DAIL-SQL, which the authors implemented already achieves 71.7 with even zero-shot and davinci-003. It seems that authors confuse the evaluation script for exec (which also requires parsing the predicted SQLs as different components https://github.com/taoyds/spider/blob/master/evaluation.py) and test-suite exec. The authors also didn't state. See how biased and confusing if authors just use one system.
    - The conclusions of Reasoning Analysis are totally not trustworthy. The big reason is that authors mistakenly assume all questions could be answered by only gold queries, which is totally not true in the text-to-SQL field. And the conclusions based on this assumption lead to confusing and misleading readers. For example, authors conclude `avg` is the hardest operation. In SQLs, you can express average aggregation by not only `avg`, but also `sum() / count()`, how did authors differentiate them? A similar thing will also happen in expressing superioist, such as the highest could be answered by `max()` and `Order By Limit 1`. Concluding only on ground-truth SQL is very unprofessional and questionable.

It seems that authors are not familiar with text-to-SQL tasks, I highly recommend removing this part instead of just misleading

1. Random Sampling is not a wise choice, in the paper [9], they said, the results will be different significantly if random multiple times. And random sampling may lead to biased settings. For example, how do you know your sampled cases randomly only contain one domain of knowledge such as medical science? In this case, your conclusions are highly biased due to your unfair distributions. We didn't see any detailed descriptions of distributions of a dataset in this paper.

**Strengths:**

1. Authors collect and conclude many tasks from existing datasets to test LCMs performance.
2. Authors post an interesting research problem settings to evaluate LCLMs on IR-based problems.

**Additional Feedback:**

none

**Clarity:**

The motivation is very not clear to me. Why did authors want to evaluate LCLMs on IR tasks? Most IR techniques are developed to be more efficient. And also according to lost-in-the-middle, feeding all contents to LLMs is not a wise and realistic choice. Also, citations in line 53. Are they hallucinated by LLMs? If so, how could I trust most contents of papers according to realistic and rigorous?

**Correctness:**

No, there is a high risk of biased settings in this dataset. See the Review part. According to the weird numbers in Table 2, the evaluation methods may be not rigorous, the authors also didn't state details in the Appendix and main contents. The experiments also contain unreasonable conclusions in text-to-SQL (Assuming only GT SQLs can answer questions accurately).

**Documentation:**

No, I cannot observe any details about this, no check for GT errors, no distributions of domains and question types.

**Ethics:**

Seems ok.

**Limitations:**

For the potential societal impact of the work, I didn't see any discussion parts in the paper. However, I think there are no problems here since the authors only collect existing datasets.

**Opportunities For Improvement:**

1. Include more recent datasets in all settings, the older dataset has the risk of data leakage problems. Authors should involve more new data at least curated by their own to illustrate to readers that their conclusions do not from data leakage significantly.
2. SQL parts should be revised significantly. Including new datasets is required (only two datasets even with the same databases now are very biased settings). Also, state clearly how authors consider the impact of the non-uniqueness of gold-SQLs to the conclusion that "`avg` is the hardest operation". This is even reversed with [8] in which authors classify questions according not only to referenced SQLs but also questions, and conclude that aggregation operations including `avg` are already good for LLMs such as GPT-4, Claude-2 even one year ago.
3. Writing should be polished. For example, why did authors change citation formatting in line 53 ([Reid et al., 2024], [Achiam et al., 2023])? Is this ChatGPT generated?
4. Authors should state motivation more clearly, why do you evaluate LCLMs for IR problems? Actually, IR tasks have their own methods to filter candidates. Take text-to-SQL as an example, all researchers only include database schema and sampled values. It's impossible to load all databases and all passages in realistic settings since most of them contain millions of rows and hundreds of pages. It's against towards rational evaluation of authors significantly.
5. A more rigorous and detailed sampling strategy is required against random sampling. and its leaded distributions of domains, and question types, should be presented. For example, authors could classify questions into different categories and sample uniformly according to each category. Otherwise, readers could not trust the settings are fair enough.
6. More open-sourced models have to be involved.

**Relation To Prior Work:**

Yes. But it seems very limited difference compared to [29] just extending the range of long contexts to 1M but this is just by high possibly random sampling and unrealistic feeding all values of db to LCLMs.

**Summary And Contributions:**

- This paper investigates the capabilities of Long-Context Language Models (LCLMs) in performing tasks traditionally handled by retrieval systems, SQL, and other Many-Shot reasoning tasks.
- They introduce LOFT, a benchmark designed to test LCLMs on tasks requiring context lengths up to millions of tokens.
- The findings show that while LCLMs can match specialized systems in textual, visual, and audio retrieval, they face challenges in complex reasoning tasks such as SQL queries.

---

> ### Author Rebuttal · Authors · 2024-08-18
>
> We thank the reviewer for the feedback. Please see our responses to each of the following points.
>
> 1. **No dataset contributions**
>
> As mentioned in Call for Datasets and Benchmarks Paper, submissions are welcome to include “thoughtfully designed (collections of) datasets based on previously available data.” Please check our third response to Reviewer GsgK's comment for more details.
>
> 2. **how such LLMs can deal with the temporal knowledge out-of-date problems?**
>
> In the LCLM paradigm, since we put the entire corpus in the prompt, one just changes the prompt to ensure that the most relevant documents are present; chain-of-thought reasoning can be used to reason the freshness of knowledge. Traditional dual-encoders have more issues with temporal problems – the index must be periodically refreshed, and the models might have to be retrained if the corpus or query distribution shifts too much.
>
> 3. **it's still the risk that LCLMs already memorize**
>
> It's important to note that the MSMARCO version used in the LOFT paper is a significantly reduced subset of the full dataset. As explained in Section 2, we subsampled the dataset to ensure the retrieval corpus fits within the LCLM context window (e.g., 128K). This version differs substantially from the MSMARCO leaderboard, making comparisons invalid. On this version, baselines like Gecko, untrained on MSMARCO, achieve high scores (Table 2), demonstrating that high performance doesn't necessarily imply LLM leakage.
> Our ablations in Tables 3 & 4 further studied the memorization effects. Table 3 shows a clear performance drop in close-book QA compared to open-book, suggesting models don’t just rely on memorization. Table 4's Titles Only Ablation reveals that models need to read the content, not just rely on memorization.
>
> 4. **Take text-to-SQL as an example, all researchers only include database schema and sampled values**,
>  **The authors choose to use execution accuracy as a metric**
>
> It is important to note that unlike traditional text-to-SQL, the LOFT SQL task lets the model generate the final answer directly from the question and the DB. It does not use any semantic parsing or SQL execution. The goal of this setup is to evaluate the inherent reasoning capabilities of LCLMs. Hence, only providing the schema wouldn’t work as the output is not the parsed query, but the final answer.
>
> As a result, GPT-4o result is not comparable to Davinci-003 DAIL-SQL. In our setup, GPT-4o must ingest the entire DB, reason in natural language, and produce the final answer. It does not use SQL at all and the task is much more challenging. DAIL-SQL is a traditional semantic parsing system that generates a SQL query which is then executed. The fact that GPT-4o is worse highlights the headrooms of LCLMs in end-to-end reasoning and math, not flawed evaluation.
>
> 5. **a very unrealistic setting for SQL**,
> **impossible to load all databases and all passages in realistic settings**,
> **Why … evaluate LCLMs on IR tasks?**
>
> It's true that LOFT's setup, constrained by current LCLM context window sizes, is limited to relatively small datasets. Traditional IR systems and semantic parsers are undoubtedly faster and less error prone. However, this criticism misses the core motivation behind LOFT.
> Our primary goal is to evaluate the inherent capabilities of LCLMs in retrieving and reasoning over large amounts of data, not to solve IR or SQL problems. We intentionally didn’t use retrievers or text-to-SQL pipelines, pushing LCLMs to handle these tasks end-to-end. The retrieval task tests how well LCLMs can locate relevant information from a large, noisy context. The SQL task tests LCLM’s inherent reasoning and math capabilities. LCLMs indeed have large headrooms in these areas.
> In practice, we don’t expect LCLMs to surpass highly specialized databases and IR systems anytime soon. However, we see a potential paradigm shift from LCLMs. For instance, not every IR task has been this optimized (e.g., new domains); instead of spending huge efforts training a new retriever, LCLMs could allow people to easily retrieve through prompting. Please also check our first response to Reviewer LA2k’s comment about SQL.
>
> 6. **according to lost-in-the-middle, feeding all contents to LLMs is not a wise**
>
> We view the "lost-in-the-middle" issue as a current model limitation, not a fundamental LCLM flaw. Future models may overcome this entirely, as Gemini 1.5 Pro already shows near-perfect performance on NIAH (https://arxiv.org/pdf/2403.05530 Figure 1). Even on LOFT, we find that LCLMs can rival traditional IR models (Table 2), and the lost-in-the-middle effect is not that much (Figure 4). LOFT's primary goal is to benchmark LCLM capabilities and identify improvement areas, not to solve IR/SQL problems definitively.
>
> 7. **mistakenly assume all questions could be answered by only gold queries**, **In SQLs, you can express average aggregation by …**
>
> As a reminder, our LCLMs do not parse to SQL. Our method does not make any function calls or any SQL execution at all. Regardless of how an average is expressed, the models tested on LOFT must do it in context in natural language. The average operator, as in the mathematical function, seemed difficult for the LCLMs. We have little awareness of the internal representation of the mathematical concept of average inside the LCLMs. We use the gold queries as a proxy for this analysis, but admittedly, it is imperfect, and we will try to clarify in the final version.
>
> 8. **... citations in line 53. Are they hallucinated by LLMs?**
>
> Thank you for pointing out our formatting error. This was simply an honest mistake caused by multiple people simultaneously editing the paper. It will be fixed in the final version.
>
> 9. **The numbers of the paper are not very clear. … that two text-to-SQL datasets can share the same length of the inputs.**
>
> Spider and SparC rely on the same corpora of databases; the queries are different but the SQL datasets share the same input lengths.

---

> > ### Author Response · Authors · 2024-08-21
> >
> > We'd like to kindly request you to take a look at our rebuttal and provide your updated review. Thank you for providing valuable feedback on our paper.

---

> > ### Comment · Reviewer_VBfb · 2024-08-30
> >
> > Thanks for the response from the authors. I'd like to retain the current score for now, as some major concerns are related to recent datasets, the SQL parts, and more open-source models that have not been fully discussed/responded to.

---

### Official Review · Reviewer_3QHy · 2024-07-25
**Review of Reviewer 3QHy**

**Rating:** 7
**Confidence:** 3
**Correctness:** The data construction seems reasonabl…
**Clarity:** The paper is well-written and it is e…

**Review:**

Overall, I enjoyed reading the paper with its novel and ambitious motivation of framing the task. A few suggestions are made in the Opportunities For Improvement section.

**Strengths:**

A novel, ambitious approach to solving various retrieval tasks using long context language models. If LCLMs become cheaper and more performant, this kind of task formulation can be the next paradigm for tasks involving retrieval.

**Additional Feedback:**

N/A

**Documentation:**

N/A

**Limitations:**

- The model transparency and the reproducibility of the exact results should also be stated. The LCLM papers that the authors cited are all works from big companies like OpenAI, Google, and Anthropic, where the rest of the world hardly knows the model recipes to replicate and can only rely on what the companies provide as APIs (which may change according to the company policies). At this moment, prompt engineering is the only option for most ML researchers to participate in this task, as none of the existing open-source LCLMs, as far as I know, are available yet to handle this scale of task in multiple modalities.
- I would suggest that the authors mention the estimated cost of running this benchmark.

**Opportunities For Improvement:**

- What is the cost of running experiments presented in this paper? All the LCLM baselines shown in this paper are API-based and I assume this will incur a significant amount of money to reproduce the result.
- Are the authors willing to release the exact input and output of Gemini and GPT-4o experiments? This would be very helpful for researchers interested in this area to start their research without spending many dollars and much time replicating the results.

**Relation To Prior Work:**

Yes, the relation to prior works is properly cited.

**Summary And Contributions:**

This paper proposes a new collection of six tasks (over 35 datasets) designed to evaluate long-context language models (LCLMs) requiring context up to 1M tokens. The authors claim LCLMs can offer a streamlined end-to-end approach and avoid error propagation for models involved in any retrieval process. The experimental results using two LCLMs including Gemini-1.5 Pro and GPT-4o show that these models outperform or are on par with traditional retrieval-based models, except for the SQL-like reasoning task. This benchmark has the potential to open a new paradigm in modeling tasks involving retrievals in the near future.

---

> ### Author Rebuttal · Authors · 2024-08-17
>
> Thank you for your constructive comments. Please see our responses below for your concerns.
>
> 1. **What is the cost of running experiments presented in this paper?**
>
> The entire LOFT 128k test sets contain around 35 datasets × 100 prompts × 128k tokens = 448M input tokens, which cost 1,568 USD for Gemini 1.5 Pro and  2,240 USD for
> GPT-4o at the time of writing. To reduce costs, we also release dev sets, which are 10x smaller and can be evaluated with around 200 USD. We expect smaller labs primarily to iterate and release on the dev set, and also for LCLMs cost per token to continue to decrease with new techniques. We will include this detail in our limitation section.
>
> 2. **Are the authors willing to release the exact input and output of Gemini and GPT-4o experiments?**
>
> We will release the precise inputs and outputs used for these models. We have already made available inputs and outputs for some datasets on https://github.com/google-deepmind/loft and are currently undergoing legal review to release the remaining datasets.
>
> 3. **The model transparency and the reproducibility of the exact results should also be stated. …  none of the existing open-source LCLMs, as far as I know, are available yet to handle this scale of task in multiple modalities.**
>
> At the time of writing, we believe there were no such open-source LCLMs. We will provide a full script to run the inference and evaluation of these API-based models so that everyone can easily reproduce the results. We already released the evaluation code and sample outputs, which users can easily run to validate their results.
>
> While API-based models were the first very long context models with multi modality support, we expect open source variants to follow (e.g.,  two weeks after we submit the paper, Llama 3.1 announced to support 128k context length), so we don’t see this as a limitation of our dataset.

---

> > ### Author Response · Authors · 2024-08-21
> >
> > We'd like to kindly request you to take a look at our rebuttal and provide your updated review. Thank you for providing valuable feedback on our paper.

---

> > ### Comment · Reviewer_3QHy · 2024-08-24
> > **Official Comment by Reviewer 3QHy**
> >
> > Thank you for the response. I look forward to the release of the dataset.

---

### Official Review · Reviewer_GsgK · 2024-08-05
**Long context evaluation**

**Rating:** 5
**Confidence:** 4
**Clarity:** yes

**Review:**

Thank you for submitting your paper to NeurIPS.

The authors mention: "The LOFT benchmark opens up a novel line of research on long-context prompting, which we introduce as Corpus-in-Context (CiC) Prompting (Section 3)" Having read section 3, I'm still quite confused on what exactly is "long-context prompting" and how is it different from just "in-context learning" and adding all the text within the context?

1M is already in the market today, a suggestion would be to expand this to ~8M to make it more future-proof atleast in the near-to-short term.

Other than Big Bench Hard (BBH) and LongICLBench which had to be re-purposed, I am under the impression the rest of the data-sets were used as is (post filtering for harmful content)? In which case, I wonder if the contributions (back filling context length) are significant enough to weigh an independent paper?
- As a subnote, please expand on what exactly was the technique of filtering:  "was done using a classifier as well as a keyword-based filtering"

**Strengths:**

The authors provide clear rationale on why the picked the tasks they did within the LOFT dataset.

Having 32K, 128K, 200K and 1M context-length samples helps evaluate the benchmark on smaller models as well.

**Additional Feedback:**

I would encourage you to also include examples that highlight the benefit of LOFT when used to eval 1M LCLMs like Gemini.

**Correctness:**

After having read the appendix, I do think the technique makes sense but the dataset was NOT submitted anywhere - so I could not look at it to comment.

**Documentation:**

No, see reference above on techniques by which the data was filtered for harmful content.

**Limitations:**

Mentioned in the review above.

**Opportunities For Improvement:**

It might help to demonstrate, why NIAH and Multi-hop QA do not bode well for evaluating LCLLMs with clear evaluation metrics which will also boost the significance of LOFT!

**Relation To Prior Work:**

Yes

**Summary And Contributions:**

The authors present LOFT. Arguing that synthetic tasks such as Needle-in-A-Haystack (NIAH) retrieval and multi-hop QA are not ideal for the extensive scaling of LCLM, Loft is a collection of 6 types of tasks, 4 modalities, and 35 datasets. For each dataset in all tasks, the authors sample up to 100 test queries, 5 few-shot queries, and 10 development queries. Further to support different models, the authors also present variations of LOFT with 32k, 128k, 200k, and 1M context length.

---

> ### Author Rebuttal · Authors · 2024-08-17
>
> Thank you for your constructive comments. Please see our responses below for your concerns.
>
> 1. **I'm still quite confused on what exactly is "long-context prompting" and how is it different from just "in-context learning" and adding all the text within the context?**
>
> Unlike standard in-context learning, which relies solely on short instructions and few-shot examples, our Corpus-in-Context (CiC) prompting emphasizes how to put a large corpus into the prompt for effective retrieval and reasoning. As shown in Table 4, vanilla prompting failed to give good quality on LOFT tasks, and each proposed component in CiC, such as shared corpus, reference-based few-shot examples, and ID echoing, significantly enhances performance. We thank the reviewer for recognizing the value of our approach, as detailed in the appendix, which we will incorporate into the main paper for the final version.
>
>
> 2. **1M is already in the market today, a suggestion would be to expand this to ~8M to make it more future-proof at least in the near-to-short term.**
>
> Thank you for the suggestion. We have already created 10M, 100M, and 1B versions of LOFT, and we will release them in our repository soon as suggested. We’ll also include more detailed findings on these larger versions in the final version of the paper.
>
>
> 3. **the rest of the data-sets were used as is (post filtering for harmful content)? In which case, I wonder if the contributions (back filling context length) are significant enough to weigh an independent paper?**
>
> As mentioned in Call for Datasets and Benchmarks Paper (https://neurips.cc/Conferences/2023/CallForDatasetsBenchmarks) submission are welcome to include “ thoughtfully designed (collections of) datasets based on previously available data.”.
>
> We did not just use the datasets in their original form. We believe these are the primary new features of LOFT:
>
> 1) A dynamic and adjustable corpus size to accommodate LCLMs of different context lengths.
> 2) Careful selection of datasets based on manual inspection to measure different LCLM skills.
> 3) Content filtering to address the somewhat unique problem of LCLMs' refusal to answer certain queries.
> 4) Corpus-in-context prompting, that requires formatting the datasets and presenting the tasks in novel ways
> 5) A new formulation of SQL that requires execution in context, measuring LCLMs’ fundamental math and reasoning abilities.
> 6) Combining datasets across four different modalities that are typically not jointly evaluated on.
>
> It's important to note that previous work like BEIR (Thakur et al, NeurIPS 2021), GLUE (Wang et al, ICLR 2019), and MTEB (Muennighoff et al, EACL 2023) are all simple collections of pre-existing datasets that require quite minimal processing, but still have driven great progress in the field.
>
> Despite the popularity of our chosen datasets, no equally robust evaluation of LCLMs existed prior to our work. Consequently, the performance of LCLMs on these datasets, especially in comparison to specialized models, was unknown. LOFT is the first to simultaneously evaluate retrieval, reasoning, and multi-modality, a task previously unattainable without LCLMs. In particular, we push on LCLMs ability to reason over structured (databases), semi-structured (retrieval corpora) and multimodal inputs in a way that can measure LCLMs abilities to do tasks that previous types of models are incapable of.
>
>
> 4. **please expand on what exactly was the technique of filtering**
>
> We removed passages containing unsafe content; this is done with a proprietary classifier that cannot be open sourced, but we have released the filtered datasets.  We also removed any content that was rejected by LCLM APIs (e.g., GPT-4, Gemini-1.5 pro) due to their content filtering. We’ll add these details about our data filtering technique in the appendix.
>
>
> 5. **why NIAH and Multi-hop QA do not bode well for evaluating LCLLMs with clear evaluation metrics which will also boost the significance of LOFT**
>
> We observed that recent LCLMs, such as Gemini 1.5 Pro (https://arxiv.org/pdf/2403.05530 Figure 1), often achieve near-perfect performance on NIAH since the tasks are synthetic. However, we aimed to demonstrate that these models still face challenges on more realistic tasks as used in LOFT, including multi-hop QA tasks like Musique. It's important to note that we never intended to suggest that multi-hop QA is not a suitable metric for evaluating LCLMs. We will clarify this point further in our updated paper.
>
>
> 6. **the dataset was NOT submitted anywhere**
>
> While our dataset was initially omitted from the paper for anonymity purposes, we've found that full anonymization isn't necessary for this track. Therefore, we're sharing it with the reviewers: https://github.com/google-deepmind/loft

---

> > ### Author Response · Authors · 2024-08-21
> >
> > We'd like to kindly request you to take a look at our rebuttal and provide your updated review. Thank you for providing valuable feedback on our paper.

---

> > > ### Comment · Reviewer_GsgK · 2024-08-27
> > >
> > > Thank you for the rebuttal. It would be interesting to see how the models perform on the extended context lengths! (10M). I have updated my score after reading the rebuttal.

---

### Author Rebuttal · Authors · 2024-08-19

We thank all the reviewers for their constructive comments and suggestions. We hope our detailed response to each reviewer’s comments addresses your concerns. In this general response, we also provide our response to some of the common concerns.

1. **Are the LOFT IR and SQL tasks realistic?**

Even though we emphasized in the paper that LOFT is a more realistic eval benchmark compared to the existing synthetically created datasets (e.g. needle-in-a-haystack), we do not intend to claim that LCLMs can replace existing specialized models. Given currently supported context lengths, inference speeds, and costs, we agree that putting entire corpora or databases into the context is not a realistic solution for IR or SQL tasks.

Nonetheless, we do think in-context retrieval and complex mathematical reasoning are necessary skills for LCLMs to be useful. We believe these tasks measure useful skills for LCLMs to have, which they could use to do tasks that no specialized models exist for, such as complex multimodal data analysis. The LOFT benchmark aims to measure the *inherent current capabilities of LCLMs* on using a given corpus or database.

In our final version of the paper, we will clarify the goals of LOFT and ensure we emphasize that *LOFT is a measurement tool for LCLMs*, rather than claiming that LCLMs are a replacement for specialized models.

2. **Confusion around LOFT-SQL**

In LOFT-SQL, we directly provide each natural language query along with the associated database in the context of an LCLM. LCLMs are asked to generate the answers directly and there are no SQL queries generated or executed. Please check our response to Reviewer LA2k for an example. While this is not a realistic solution for the SQL task, it provides a novel way to evaluate LCLM reasoning headroom, as a highly capable LCLM should be able to consume a database in its context (whenever it fits, which is the case for our chosen public datasets) and reason over it to provide correct answers for the natural language queries. In the final version of the paper, we will emphasize that LOFT-SQL is a testbed for measuring the capabilities of LCLMs to do complex reasoning, but not a viable alternative to existing pipelined systems that use a SQL executor.

3. **Isn’t LOFT merely a group of existing datasets?**

As mentioned in Call for Datasets and Benchmarks Paper (https://neurips.cc/Conferences/2023/CallForDatasetsBenchmarks), submission are welcome to include “ thoughtfully designed (collections of) datasets based on previously available data.”

We did not just use the datasets in their original form. We believe these are the primary new features of LOFT:

1) A dynamic and adjustable corpus size to accommodate LCLMs of different context lengths.

2) Careful selection of datasets based on manual inspection to measure different LCLM skills.

3) Content filtering to address the somewhat unique problem of LCLMs' refusal to answer certain queries.

4) Corpus-in-context prompting, that requires formatting the datasets and presenting the tasks in novel ways.

5) A new formulation of SQL that requires execution in context, measuring LCLMs’ fundamental math and reasoning abilities.

6) Combining datasets across four different modalities that are typically not jointly evaluated on.

It's important to note that previous work like BEIR (Thakur et al, NeurIPS 2021), GLUE (Wang et al, ICLR 2019), and MTEB (Muennighoff et al, EACL 2023) are all simple collections of pre-existing datasets that require quite minimal processing, but still have driven great progress in the field.

4. **How can I get access to the dataset?**

Our dataset was initially omitted from the paper for anonymity purposes. As we've found that full anonymization isn't necessary for this track, we're sharing it with the reviewers: https://github.com/google-deepmind/loft

---

### Decision · Program_Chairs · 2024-09-26

**Decision:**

Reject

**Comment:**

This paper has received ratings ranging from 2 to 7, and the authors have made significant efforts to address the reviewers' concerns. The work involved in collecting such a large dataset, organizing the tasks, and conducting experiments is substantial. However, the confusion between 'in-context learning' and 'long-context prompting' (and the associated Corpus-in-Context prompting) suggests that the new task setting is less convincing and difficult to accept. The main weakness lies in the SQL part, which is hard to justify. While the authors argue that the model is powerful enough to bypass SQL generation and produce results directly from structured data, the reviewers consider this setting unrealistic for fitting entire databases into models as context, limiting the models' applicability to small relational databases. Overall, much more effort is needed to make the paper more convincing, particularly regarding long-context prompting and the task settings. My recommendation, based on the paper and the comments, would be to completely remove the SQL part to avoid confusion.